


# The Global Ozone Monitoring Experiment: Review of in-flight performance and new reprocessed 1995–2011 level 1 product

Melanie Coldewey-Egbers[1], Sander Slijkhuis[1], Bernd Aberle[1], Diego Loyola[1], and Angelika Dehn[2]

[1]German Aerospace Center (DLR), Remote Sensing Technology Institute, Oberpfaffenhofen, Germany
[2]European Space Agency (ESA), ESRIN, Frascati, Italy

**Correspondence:** Melanie Coldewey-Egbers (Melanie.Coldewey-Egbers@dlr.de)

**Abstract.** The Global Ozone Monitoring Experiment (GOME) on-board the second European Remote Sensing satellite provided measurements of atmospheric constituents such as ozone or other trace gases for the 16 year period from 1995 to 2011. In this paper we present a detailed analysis of the long-term performance of the sensor and introduce the new homogenized and fully calibrated level 1 product which has been generated using the recently developed GOME Data Processor level-0-to-1b (GDP-L1) Version 5.1. By means of the various in-flight calibration parameters we monitor the behavior and stability of the instrument during the entire mission. Severe degradation of the optical components has led to a significant decrease in intensity in particular in channels 1 and 2 covering the spectral ranges of 240–316 nm and 311–405 nm, respectively. Thus, a soft correction based on using the sun as a stable calibration source is applied. Revision and optimization of other calibration algorithms such as the wavelength assignment, polarization correction, or dark current correction resulted in an improved and homogeneous level 1 product that can be regarded as the European satellite reference data for successor atmospheric composition sensors and that provides an excellent prerequisite for further exploitation of GOME measurements.

## 1 Introduction

The Global Ozone Monitoring Experiment (GOME) was launched on 21 April 1995 by the European Space Agency (ESA) on-board the second European Remote Sensing satellite (ERS-2). It was the first European UV-VIS-NIR (ultraviolet–visible–near-infrared) spectrometer in space dedicated to observe atmospheric trace constituents such as ozone or nitrogen dioxide as well as cloud and aerosol parameters on a global scale (Burrows et al., 1999). The sensor operated for more than 16 years, which is a world record for this kind of instruments, until the retirement of the ERS-2 platform in early July 2011. GOME is the predecessor of a series of similar follow-up instruments like SCIAMACHY (Scanning Imaging Absorption Spectrometer for Atmospheric Chartography, 2002-2012, Bovensmann et al., 1999) on-board Envisat, OMI (Ozone Monitoring Instrument, launched in 2004 on-board Aura, Levelt et al., 2006), or GOME-2 (Munro et al., 2016), and marks the beginning of European operational, global, long-term monitoring of climate-relevant atmospheric parameters.

The existing atmospheric data archive of GOME is of very high value and may be considered (in conjunction with SCIA-MACHY) as the basis for a future reference data set for successor sensors. The status of scientific results of the miscellaneous GOME level 2 data products is presented in numerous publications (e.g., Balis et al., 2007; de Smedt et al., 2008; Loyola





et al., 2010; van Roozendael et al., 2012; Lerot et al., 2014). Furthermore, GOME data form a substantial part of recently developed long-term climate data records, for example the GOME-type Total Ozone Essential Climate Variable and the harmonized tropical tropospheric ozone data records generated within the framework of ESA's outstanding Climate Change Initiative (Coldewey-Egbers et al., 2015; Heue et al., 2016). In 2012 at ESA's Atmospheric Science Conference, as a result of the discussion rounds, the scientific user community formulated a set of recommendations (ATMOS, 2012) also addressing the preservation and further exploitation of the 16 years of GOME measurements. These recommendations have led to ESA's GOME-Evolution project that started in April 2014. Among other topics, the objective of this activity is to provide the Earth Observation (EO) user community with improved and consolidated GOME level 1 products, in an easily accessible common data format, based on updated GOME calibration algorithms and improved in-flight calibration characterization for the complete mission. Homogenization of the current GOME level 1 products has become necessary because so far they were generated using different processor versions and, thus, were not fully consistent during the complete mission.

Furthermore, a detailed investigation of the long-term performance of the GOME instrument for the entire mission period was carried out in the framework of GOME-Evolution. The results will be presented in this paper. This part of the study is an extension to the work by Coldewey-Egbers et al. (2008) who introduced a first overview of the long-term behavior for the 11 year time span from 1995 to 2006. Special emphasis is put on the analysis of the Sun mean reference spectra in order to monitor and correct for the gradual degradation of the instrument's optical properties. Coldewey-Egbers et al. (2008) have shown that the degradation is severe in particular in the UV channels 1 ($\sim$70-90%) and 2 ($\sim$35-65%) covering the spectral range of 240–316 nm and 311–405 nm, respectively. Similar changes are observed for SCIAMACHY and GOME-2 (Noël et al., 2007; Bramstedt et al., 2009; Munro et al., 2016), whereas it is considerably smaller for OMI (Schenkeveld et al., 2017).

The various in-flight calibration parameters are a good means to monitor the long-term stability of the sensor and its measurements. Instrument stability is one of the most important prerequisites to meet the challenge of measuring very small changes in atmospheric parameters associated with long-term climate change from space. For example, satellite sensors are required to detect ozone trends in the order of 1% decade$^{-1}$ (GCOS, 2011). Amongst other things, in our study particular attention was paid to the analysis of the long-term performance and stability of the spectral calibration since errors in wavelength assignment may have a significant impact on the earth albedo and trace gas retrievals (Voors et al., 2006; van Geffen et al., 2015; Pan et al., 2017).

In addition to the new GOME level 1 product and the revised in-flight calibration data set, a "Climate" total column water vapor product has been developed within ESA GOME-Evolution (Beirle et al., 2018). It is based on homogenized GOME, SCIAMACHY, and GOME-2 observations and provides a consistent time series that is dedicated to study the temporal evolution of water vapor over the past two decades on a global scale. Another part of the project was the creation of a web gallery (GOME Web Gallery) featuring the GOME/ERS-2 mission and related scientific achievements.

The paper is organized as follows: In Section 2 we provide an overview of the GOME instrument design (Sect. 2.1), a brief description of the level-0-to-1 algorithms and processing chain (Sect. 2.2), a summary of the calibration algorithms using results of the on-ground instrument characterization (Sect. 2.3), and a short decription of the new level 1 product generated in the framework of GOME-Evolution (Sect. 2.4). An analysis of the Sun mean reference spectra and the description of the degrada-





tion correction algorithm is presented in Sect. 3.1, followed by the investigation of the Polarization Measurement Device data (Sect. 3.2). In Section 3.3 we show results of the reflectance degradation analysis. Section 4 contains the detailed results of the long-term analysis of the most important GOME in-flight calibration parameters needed for, e.g., the spectral calibration or the dark current correction. Summary and concluding remarks are finally given in Section 5.

## 2 GOME/ERS-2

### 2.1 Instrument and platform characteristics

GOME is a nadir-viewing, across-track scanning spectrometer that covers the ultraviolet, visible and near-infrared wavelength range from 240 to 790 nm with moderate spectral resolution of 0.2 to 0.4 nm. It measures the solar radiation reflected and scattered by the Earth's atmosphere and surface as well as the solar irradiance. Its primary objective is the determination of the amounts and distributions of atmospheric trace constituents, such as ozone, nitrogen dioxide, sulfur dioxide, formaldehyde, or bromine oxide as well as cloud and aerosol parameters (Burrows et al., 1999). In normal viewing mode, there are three forward scans (footprint size of $320 \times 40 \, \text{km}^2$ each – across-track $\times$ along-track) followed by a backscan with 1.5 s integration time. The maximum swath width is 960 km, global coverage is achieved at the equator within three days.

GOME is a double monochromator, which has as dispersing elements a predisperser prism, combined with a holographic grating in each of the four optical channels. The earthshine radiance and solar irradiance spectra are recorded with four linear Si-diode arrays with 1024 spectral elements each. These detectors are cooled to 235 K by means of Peltier coolers to reduce dark current and to improve the signal-to-noise ratio. The four channels cover the wavelength regions of 240–316 nm (channel 1), 311–405 nm (channel 2), 405–611 nm (channel 3), and 595–793 nm (channel 4). Channels 1 and 2 are further electronically divided into two bands ('a' and 'b') covering the short-wavelength and long-wavelength parts of the channels, respectively. In addition there are four stray light bands: two short-wave of band 1a, one long-wave of band 1b, and one short-wave of band 2a. Part of the light is branched out at the predisperser prism and recorded with three fast broadband silicon photo-diodes, the polarization measurement devices (PMDs), whose spectral ranges cover approximately the optical channels 2 (300–400 nm), 3 (400–580 nm), and 4 (580–750 nm), respectively. They measure the amount of light polarized parallel to the instrument slit, which is perpendicular to the plane of incidence of the scan mirror. The PMDs are non-integrating detectors which are continuously sampled, albeit over an RC-circuit which has an averaging effect over the sampling time. The corresponding sampling time of the PMD measurements is 93.75 ms, i.e. 16 PMD measurements are available for one detector channel measurement at the default integration time of 1.5 s

The various pointing geometries of the GOME scan mirror permit in addition to solar and earth nadir viewing, polar viewing (viewing angle of 45°), and lunar observations (viewing angle of about 80°) at selected times during a year. A calibration unit adjacent to the spectrometer part consists of the sun view port and a compartment housing a platinum-neon-chromium (Pt/Ne/Cr) hollow cathode discharge lamp. The solar radiation is attenuated by a 20% transmission mesh and directed via a diffuser plate (wet-sanded aluminium plate with chromium/aluminium coating) onto the entrance slit of the spectrometer. The calibration unit becomes optically coupled to the spectrometer by appropriate positioning of the scan mirror.





A detailed overview of the GOME instrument, its operation, and scientific methods can be found in the GOME User's Manual (GOME Users Manual) and in Burrows et al. (1999). For understanding the algorithm principles described in the following sections a simple functional diagram of the GOME instrument is shown in Fig. 1. The most important instrument components relevant to the level-0-to-1 calibration are

– the scan mirror whose position is linked to the observation mode, e.g., nadir or pole scanning, static or moon view, and the calibration mode; the latter comprises solar measurements, dark signal measurements, and spectral lamp measurements;

– the calibration unit that hosts the spectral calibration lamp and the Sun diffuser;

– the slit that limits the instantaneous field-of-view to $2.9° \times 0.142°$ or $40 \times 2$ km$^2$ on the ground; the slit function, i.e., the instrument spectral response to monochromatic input, is a convolution of projected slit width, pixel response, and optical

abberations;

– the quartz predisperser prism where part of the light is branched out and directed towards the PMD unit (see below);

– the channel separator that separates the wavelengths for channel 1, for channel 2, and for channels 3 and 4, respectively; this separation serves to reduce stray light on the UV detectors, i.e. channels 1 and 2;

– the dichroic filter that separates the wavelengths of channel 3 from those of channel 4; in the spectral range from 590 nm

to 610 nm the filter changes from reflection to transmission;

– the channel optics that consists of 4 quartz lenses mounted in one barrel;

– four red LEDs which illuminate the detectors directly and which are used to characterize the pixel-to-pixel sensitivity;

– the Focal Plane Assembly (FPA) which holds the Reticon detector and the pre-amplifier electronics;

– the PMD unit that contains three broadband polarization measurement devices whose spectral bandwidths correspond

roughly to the detector array channels 2, 3, and 4.

ERS-2 orbited the Earth at an altitude of about 790 km in a Sun-synchronous near polar orbit; the descending node local equator crossing time was about 10:30 UTC, and a repeat cycle of 35 days. Each orbit takes ∼100 min and the spacecraft completes ∼14 orbits per day. Operational GOME observations are available since July 1995, although global coverage was lost in June 2003 due to a permanent failure of the ERS-2 on-board tape recorder. Since then availability of GOME data

coverage is limited to the region where ERS-2 was in direct contact with ground stations in the European-Atlantic sector. Over the years additional ground stations have been brought online to incrementally increase the data gathering abilities of the satellite. The ERS-2 active mission was completed on 4 July 2011 at orbit no. 84719. The ESA mission operations overview (GOME Mission Operations Overview) provides a detailed review of the most important events over the entire mission lifetime which may have had an impact on the GOME data quality. Anomalies such as cooler or instrument switch-offs, spectral lamp

failures, or data gaps are reported on a yearly basis.



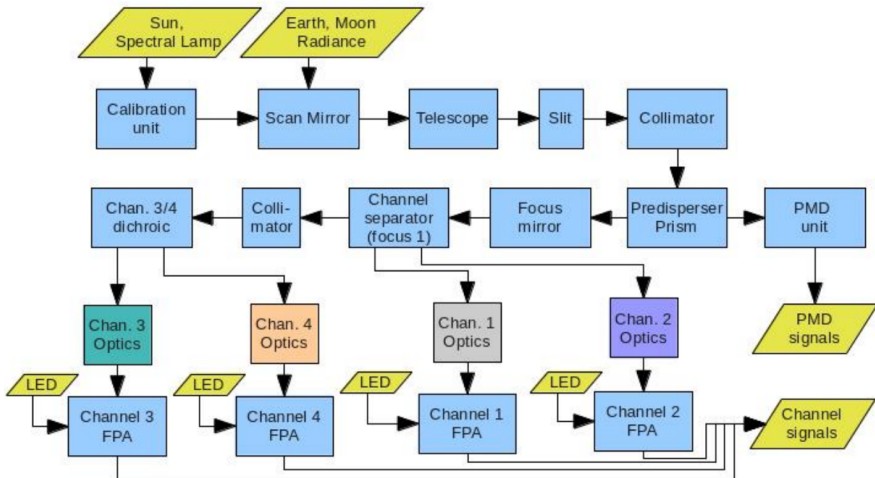

**Figure 1.** Functional diagram of the GOME instrument (see text for more explanations).

## 2.2 GOME Data Processor

The GOME Data Processor (GDP) is the operational off-line ground segment for the GOME instrument, incorporating, among other things, a level 0-to-1 processing chain (GDP-L1) and the complete GOME data archive (Loyola et al., 1997). During the level 0-to-1 processing, GOME data is converted into calibrated physical quantities by applying a series of calibration algorithms. Some of the calibration data were obtained during the pre-flight on-ground calibration. Other parameters which can be directly derived from measurements using on-board calibration sources are derived during the level 0-to-1 processing. The basic calibration steps needed are:

- signal correction, i.e. correction for dark signal, Focal Plane Assembly (FPA) crosstalk, pixel-to-pixel variations (PPG) in quantum efficiency, and stray light;

- wavelength calibration, i.e. assigning to each detector pixel its associated wavelength;

- radiance calibration, i.e. conversion of the corrected detector signals to radiance units by application of the radiance response function; this step also includes the polarization correction;

- irradiance calibration, i.e. conversion of the corrected detector signals to irradiance units, including the correction for BSDF (Bi-directional Scattering Distribution Function) of the diffuser plate;

- geolocation, i.e. determination of the geographical position for each detector readout using ESA's ERS-2 orbit propagator;

- quality assessment, i.e. identification of dead pixels, hot pixels, saturation, and sun-glint.





The GOME on-ground calibration was performed during the pre-flight calibration phase by TPD/TNO (Netherlands Organization for Applied Scientific Research). The output was a data set containing the so-called "Calibration Key Data" such as stray light correction, BSDF coefficients, radiance response function, or polarization correction. In the course of switching from on-ground to the in-flight situation various adjustments in the key data had to be applied which were mostly due to air-vacuum wavelength shifts and outgassing of optical coatings. Over the years further updates of the key data have been implemented which were related to the radiance response and to the diffuser BSDF (e.g., Aben et al., 2000; Slijkhuis et al., 2006). An overview of these algorithms using the on-ground calibration data is given in Sect. 2.3.

Calibration constants which can be directly deduced from measurements using on-board calibration sources are derived during the level 0-to-1 processing. This comprises the dark signal measurements on the night side of each orbit, and at regular intervals wavelength calibration using the spectral lamp measurements as well as PPG calibration using internal LED measurements. Monitoring these calibration parameters provides an excellent insight into the long-term stability of the instrument. A detailed description of the corresponding algorithms and the results of the long-term analysis is presented in Sect. 4.

Another step in the entire calibration procedure is the correction of degradation (see Sect. 3.1). Due to degradation in optical components the calibration parameters for radiance and irradiance change in time. However, this degradation cannot be derived from on-board calibration sources and the correction has to be obtained offline and externally from the data processor. For GOME this has been done by scientific analysis of the solar observations.

The last major GDP-L1 processor update has been developed in 2006 in order to provide a first complete reprocessing of the data set available at that time. The main driver for this activity has been the gaps in the solar calibration spectra over long periods caused by pointing issues on the ERS-2 platform. Furthermore, other algorithmic developments have been included and a detailed analysis of the long-term performance of GOME in terms of numerous diagnostic in-flight calibration parameters has been performed for the first 11 year period (Coldewey-Egbers et al., 2008).

In the framework of ESA's GOME-Evolution project, the GDP-L1 Version 5.1 has been developed in order to generate a completely homogenized, fully calibrated level 1 product for the entire 16 year mission period. Algorithm improvements comprise a new polarization correction (Sect. 2.3.4) and an updated degradation correction (Sect. 3.1), an improved usage of dark signal measurements (Sect. 4.3), as well as revised and improved spectral calibration (Sect. 4.2).

## 2.3 GOME on-ground calibration data and correction algorithms

In this section we provide an overview of the GOME on-ground calibration data and the basic principles of the corresponding correction algorithms. For more details we refer to Slijkhuis and Aberle (2016).

### 2.3.1 Correction for FPA noise and band 1a residual offset

An additional source of temporary slow noise on the Reticon detector signal is crosstalk correlated to the voltage controlling the Peltier coolers on the Focal Plane Assembly. It can be approximated by multiplying the Peltier cooler control signal by a scaling factor which has been obtained from one typical orbit and is stored in the calibration key data file. The noise is correlated to the integration time and correction is only necessary for integration times of 6 s or longer typical for band 1a

measurements. Furthermore, the correction is only applied to Earthshine measurements. The correction algorithm comprises four steps: (i) apply a high-pass filter to all Peltier output signals from one orbit, (ii) calculate an average value of the filtered Peltier output, (iii) multiply the mean Peltier output by the scaling factor specified for the actual integration time, and (iv) subtract the noise from the signals of the entire band to be corrected.

It appears that after the removal of the Peltier noise as described above, a residual offset remains which has to be corrected since it is too large for, e.g. ozone profile retrieval. This additional correction has been implemented in the framework of the CHEOPS-GOME study (Slijkhuis, 2006; Slijkhuis et al., 2006) and uses the signal of the stray light band 1a (just before the beginning of the nominal band 1a).

### 2.3.2   Stray light correction

After the first calibration and characterization measurements of GOME at TPD/TNO, it became obvious that stray light, i.e. light from wavelengths other than the nominal wavelength of a specified detector pixel, is a major issue and needs to be corrected during the in-flight calibration exercise. Specifically in channels 1 and 2 the signal readouts are spoiled by a non-negligible amount of stray light whose main sources are:

- a uniform or very slowly changing quantity of stray light over the detector pixels induced by diffuse reflections within
the FPA;

- ghost stray light signals induced by reflections from the surfaces of the detector arrays and the lenses of the channel telescope; symmetrical ghosts (signals mirrored at the middle of the detector) and asymmetrical ghosts (signals mirrored at some arbitrary detector pixel) were detected;

- out-of-band stray light on the PMDs induced by radiation outside the wavelength range of the detector arrays.

In order to reduce the impact of stray light several improvements were applied before launch such as tilt changes to the gratings, the use of anti-reflection coatings, change of the channel separation between channel 1 and 2, and improvement of internal baffling. Despite these improvements a correction algorithm is still required. Therefore, summed contributions from uniform and ghost stray light are subtracted from the measured signal. The relative uniform stray light levels obtained during the pre-flight calibration are 0.2% for channels 1 and 2, and 0.1% for channels 3 and 4, respectively. These levels are multiplied
with the averaged signal fluxes per detector array to get the uniform stray light contribution. For GOME, currently there is only one significant ghost. Its efficiencies (0.05% for channels 1, 2, and 4 and 0.1% for channel 3) were determined during the pre-flight characterization and are multiplied with the mirrored (around the pixel center of the ghost) signal flux to get the ghost stray light contribution. However, the calibration key data for stray light are probably not more accurate than ∼10%.

### 2.3.3   Radiometric calibration

The objective of the radiometric calibration is to transform the 16-bit binary units (BU) of the detector pixel readouts into calibrated radiances (photons s$^{-1}$ cm$^{-2}$ nm$^{-1}$ sr$^{-1}$) or, for the sun, into calibrated irradiance (photons s$^{-1}$ cm$^{-2}$ nm$^{-1}$). For





earthshine measurements the intensity calibration also includes the application of a polarization correction (see Sect. 2.3.4), whereas for moon measurements a 'basic' intensity calibration is applied.

In GDP-L1 the radiometric calibration is divided into several steps. A basic calibration is carried out using the key data from the on-ground calibration. This step comprises the application of the radiance response function which depends on wavelength,
scan angle, and temperature as well as the application of a basic BSDF which is dependent on the wavelength, the azimuth angle, and the elevation of the sunlight on the diffuser. The radiance response function is a compound function in which the scan angle dependent part and the temperature dependent part are given per channel, for 9 scan angles and for 5 temperatures, respectively. These key data are then interpolated to the actual values of the respective measurement. The BSDF is expressed as parametrization using polynomials. Subsequently, refinements are made to correct for instrument degradation (see Sect. 3.1)
and to correct the solar irradiance using an improved azimuth dependence of the diffuser BSDF (Slijkhuis et al., 2006). For the latter the azimuth dependence is fitted using a third-order polynomial in wavelength for all channels. The polynomial coefficients are stored in a look-up-table for a number of azimuth angles which are then linearly interpolated to the actual azimuth angle.

Furthermore, the earthshine radiance is corrected for the so-called 'radiance jump' effect that is caused by the serial readout
of the detector, i.e. the last pixel of the array is read out 93.75 ms later than the first pixel. In case of inhomogeneous ground scenes this effect may be visible as a jump in radiance between two neighboring detectors. The last pixels of one detector record the same wavelengths as the first pixels of the next channel, but at an integration time shifted by 93.75 ms. A linear correction in wavelength is applied which re-normalizes all intensities to the same integration time thereby using information from the PMDs (which are read out every 93.75 ms synchronized with the first detector pixel). Although the correction adjusts
the continuum level, it cannot account for any difference in spectral features that may arise from viewing a slightly different ground pixel.

### 2.3.4 Polarization correction

GOME is a polarization sensitive instrument. The radiance response function described in Sect. 2.3.3 calibrates the instrument assuming unpolarized light. Therefore a correction factor must be applied which describes the ratio of the throughput for actual
input polarization to the throughput for unpolarized light. The polarization correction algorithm (PCA) needs the polarization sensitivity of the instrument as well as a characterization of the atmospheric polarization, and it is divided into two main parts. The first step is to derive the atmospheric polarization from theory and from measurement for a few wavelengths. Three of these polarization points come from the comparison of channel array signals with broadband PMD signals; the corresponding wavelengths are approximately 360 nm, 500 nm, and 700 nm. A fourth point is obtained from theoretical assumptions and
comes from a Rayleigh single-scatter model simulation of polarization in the UV. The second step of the PCA is to interpolate the polarization points to wavelength and to apply the correction to the whole spectrum. Akima interpolation is used for the better part of the spectrum longward of 325 nm. Below 300 nm polarization is taken as a constant which is calculated based on a Rayleigh single-scattering model. In the UV region 300-325 nm the Generalized Distribution Function (GDF, Schutgens and



Stammes, 2002) is used. The impact of the polarization correction on the spectra is in the order of -1±5%. The largest change (-5±12%) arises in band 1b (283–316 nm) and the minimum impact is found in channel 4 (-0.5±0.5%).

Within GOME-Evolution one important improvement for the GDF parametrization has been implemented, that is to use GOME's own retrieved total ozone columns instead of climatological ones. To this end the level 2 ozone values are inserted into

the level 1 calibration database. This was hardly possible during the operational phase of the instrument, but for reprocessing there was no limitation, especially because the ozone retrieval is not critically dependent on the polarization curve itself, i.e. within the accuracy needed for this parametrization. Thus, in principle no iterations between level 1 processing and level 2 processing are necessary. Nevertheless, in practice these iterations were made so that ozone columns used for the final version are fully compatible with the level 1 polarization.

## 2.4 New GOME level 1 product

The previous GOME level 1 data products from the predecessor GDP-L1 Version 4.x and lower contained geolocation, uncalibrated measurements, plus all necessary calibration data (and thus was in modern terminology more like a level 1a product). In addition an external post-processing software "extractor" tool was needed to convert these data to calibrated radiances, or to calibrated solar irradiance, respectively. The advantages were a small product size and the flexibility for the scientific user to

perform sensitivity studies on the impact of different calibration steps. However, in the course of time it turned out that both arguments are no longer valid.

In lieu thereof, the new GOME level 1 product generated with GDP-L1 Version 5.1 contain fully calibrated (ir)radiances, corresponding geolocation information, and selected calibration parameters in NetCDF-4 format. Running a separate extraction tool is not necessary anymore; several former extraction software options are now integrated in GDP-L1, others are no longer

used (Slijkhuis and Aberle, 2016). The product format and structure are designed to be similar to currently developed or planned EO products, in particular to the Sentinel-5 Precursor mission launched in October 2017. This should enable the application of common reading software to the different atmospheric composition sensors with little or no adaptions required for the various products. In addition to radiance and irradiance data, cloud parameters retrieved with the OCRA (Optical Cloud Recognition Algorithm) and ROCINN (Retrieval of Cloud Information using Neural Networks) algorithms (Lutz et al., 2016;

Loyola et al., 2018) have been integrated in the new level 1 product which required reprocessing of the data record in several iterations. Following the request from the users, another addition compared to the old product is geolocation information for each single PMD measurement. A more detailed description of the content and structure of the new level 1 product can be found in Appendix A and in the GOME/ERS-2 Product User Manual (Aberle, 2018).





## 3 Solar irradiance, PMD measurements, and reflectance

### 3.1 Sun mean reference spectrum and degradation correction algorithm

Once per day GOME records a short series of Sun spectra via the solar port and a diffuser plate. Thereby, the incidence angle
on the diffuser is (i) constant in azimuth (which varies only with season) and (ii) changes in elevation as the Sun moves

through the field-of-view. The incidence angle of the scan mirror is $41°$(compared to $49° \pm 15°$for the nadir measurements).
All measurements within an elevation angle of $\pm 1.5°$with respect to the center are averaged and corrected for the azimuth
dependence of the diffuser BSDF (see Sect. 2.3.3). This yields the so-called daily Sun mean reference (SMR) spectrum which
is stored in the calibration database and used for the calculation of the earthshine reflectivity spectra. The latter themselves
serve as input for almost all retrieval algorithms for atmospheric constituents as well as cloud and aerosol properties.

The relative intensity of the GOME SMR spectra with respect to a reference spectrum from 3rd July 1995 is depicted in Fig. 2
(January 1996 to January 2011, one spectrum per year) to demonstrate the severe impact of degradation of the optical properties.
This comparison shows that the pre-flight radiance parameters were no longer applicable to the in-flight situation (Aben et al.,
2000; Hegels et al., 2001). The main degradation as a consequence of extensive exposure to the space environment can be
attributed to deposits on the scan mirror (which is coated with a $MgF_2$ layer) thereby changing its reflective properties. The

loss in throughput is especially severe in channel 1. Below 300 nm intensity decreased by 80-95% which implies a significant
deterioration of the signal-to-noise ratio. Table 1 indicates the approximate signal-to-noise ratios for typical radiance values
for channel 1 at 305 nm, channel 2 above 325 nm, and channels 3 and 4, respectively, at the beginning of the GOME mission.
In general, the signal-to-noise ratio is well above 1000. Towards shorter wavelengths the ratio significantly decreases due
to strong ozone absorption and a weaker solar irradiance. These values are comparable to those obtained for SCIAMACHY

(Bovensmann et al., 1999, their Fig. 4). In channel 1 (and also in the other channels) the signal-to-noise ratio for GOME is
expected to decrease linearly with the degradation of the signal (for signal levels below $\sim$15.000 BU) since the detector noise
is exceeding the shot noise. Above signal levels of $\sim$15.000 BU shot noise becomes dominant. Thus, the strong degradation
observed in channel 1 may have a severe impact on the retrieval of atmospheric parameters using this specific spectral region,
e.g. ozone profiles (van Peet et al., 2014). The decrease in channel 2 is 40-80%. In channel 3 the decrease (10-40%) started

in 2001 when the measurements were additionally affected by an ERS-2 pointing problem as a consequence of the loss of
the gyroscope functionality. Throughput changes in channel 4 are relatively small. SCIAMACHY as well as GOME-2 suffer
from degradation in pretty much the same way (Noël et al., 2007; Bramstedt et al., 2009; Munro et al., 2016), whereas OMI
irradiances degraded by only 3-8% from 2005 to 2015 (Schenkeveld et al., 2017).

In Fig. 2 at wavelength regions around 470 nm and 600 nm changes in intensity are affected by changes (outgassing of

coatings) in the dichroic filter which separates the wavelengths of channels 3 and 4. Unpredictable polarization-sensitive
changes were observed and the radiometric calibration in these regions might be doubtful. Furthermore the outgassing is
assumed to be responsible for the slight transmission increase in channel 3 in the early part of the mission. The low-frequency
oscillating structure appearing in all channels is the result of the etalon effect which is caused by a changing thickness of ice





**Figure 2.** Relative intensity of GOME Sun mean reference spectra (January 1996 to 2011, one spectrum per year) with respect to a reference spectrum from 3rd July 1995. Corresponding smooth solid lines denote results of the polynomial fit performed during the degradation correction. Wavelength regions around 470 nm and 600 nm are affected by changes (outgassing) in the dichroic filter (see text for more details).

**Table 1.** Approximate signal-to-noise ratios at the beginning of the GOME mission.

| Spectral region | Channel 1 (at 305 nm) | Channel 2 ($\geq$325 nm) | Channel 3 | Channel 4 |
|---|---|---|---|---|
| Radiance [photons s$^{-1}$ cm$^{-2}$ nm$^{-1}$ sr$^{-1}$] | $\sim$4.0e11 | $\sim$2.0e13 | $\sim$3.0e13 | $\sim$3.0e13 |
| Signal-to-noise | $\sim$1100 | $\sim$3500 | $\sim$4000 | $\sim$2500 |

deposits on the detectors and which leads to spectral interference patterns (Mount et al., 1992). At present no attempts are being made to correct for this effect.

In order to remediate the observed GOME science channel degradation a correction algorithm was developed in the framework of the ESA project GDAQI (GOME Data Quality Improvement, Aben et al., 2000). This degradation correction is applied to irradiance and radiance spectra as additional part of the radiometric calibration. The degradation correction approach that was chosen is the comparison of all available solar data from the entire mission period with the corresponding solar data of a reference day in the early GOME lifetime (3rd July 1995). The Sun is a reliably stable input source to monitor the instrument throughput despite of small changes in the solar spectrum due to changes in solar activity. This study was done for both the GOME science channels and the PMDs. The temporal changes have been determined by building ratios of all solar spectra with the solar spectrum of the reference day $t_0$ which may be written as

$$\frac{I_{Sun}(\lambda,t)}{I_{Sun}(\lambda,t_0)} = P_{Deg}(\lambda,t) \cdot C_{SED}(t) \cdot Residual(\lambda,t), \tag{1}$$

where $P_{Deg}(\lambda,t)$ is the used degradation function dependent on wavelength $\lambda$ and time $t$. $C_{SED}(t)$ is the intensity correction due to the seasonal variation in Sun-Earth distance, and $Residual(\lambda,t)$ is the remaining structure. Note that the impact of the etalon effect and the changes in the dichroic filter are not accounted for. For the determination of the degradation correction



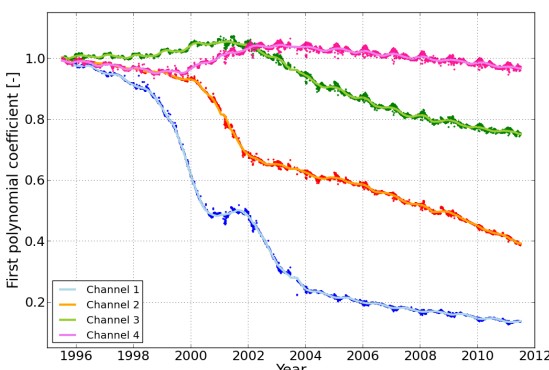

**Figure 3.** First polynomial coefficient $a_0$ from wavelength fit as a function of time (small dots, blue: channel 1, red: channel 2, green: channel 3, and magenta: channel 4). The light blue, orange, light green, and light violet curves denote the corresponding smoothed curves using a Savitzky-Golay smoothing filter with a filter width of 250 days.

function $P_{Deg}(\lambda, t)$ a two-step approach was developed: (i) each irradiance ratio (per channel) is approximated by a polynomial function in wavelength and (ii) each coefficient of this polynomial in wavelength is subsequently described by a time-dependent expression. Thus, for the degradation function $P_{Deg}(\lambda, t)$ per channel the following expression has been obtained:

$$P_{Deg}(\lambda, t) = \sum_{k=0}^{n} a_k(t) \cdot (\lambda - \lambda_0)^k. \tag{2}$$

$\lambda_0$ is the center wavelength in each channel. Each coefficient $a_k(t)$ of the polynomial in wavelength is taken from a look-up table (LUT). For channels 1 and 2 third order polynomials ($n = 3$) are used, whereas in channels 3 and 4 quadratic ($n = 2$) and linear ($n = 1$) polynomials are used, respectively (see smooth curves in Fig. 2). The LUT is generated by smoothing the time series of each polynomial wavelength coefficient using a Savitzky-Golay filter (Press et al., 1992) with a filter width of 250 days. Figure 3 shows, for each channel, the first polynomial coefficient ($a_0$) and the corresponding smoothed curve as a

function of time. In channel 1 degradation started almost immediately after launch. Until 2000 the intensity decreased to ∼50% of the early-mission values. In 2001/02 measurements were additionally affected by the ERS-2 pointing problem. In channel 2 significant decrease in intensity is observed especially during this period. Channels 3 and 4 are additionally affected by changes in the dichroic filter (outgassing of coatings).

In GDP-L1 the degradation correction is then applied according to:

$I_{SunCorr}(\lambda, t) = I_{Sun}(\lambda, t)/P_{Deg}(\lambda, t).$ \hfill (3)

In addition to the Sun the moon provides an independent irradiance source and in principle GOME lunar measurements can be used to characterize and monitor instrument performance and degradation (Dobber, 1997; Dobber et al., 1998). The moon is viewed on the eclipse side of the orbit over the scan mirror at an incidence angle of $5° - 15°$. The amount of light is of the same order of magnitude as for the earthshine observations, though calibration measurements are complicated by several





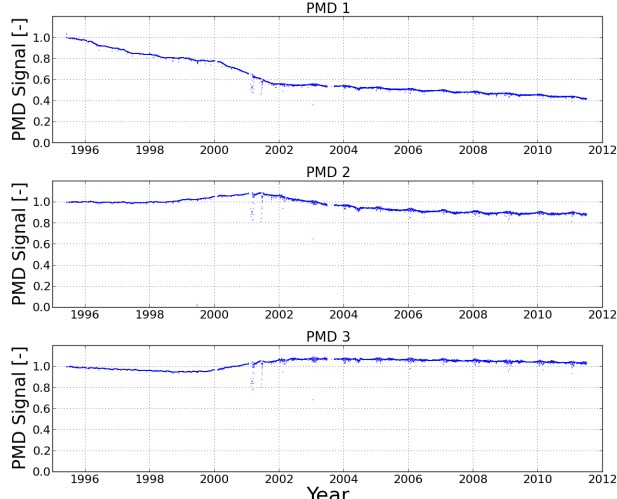

**Figure 4.** Relative change of PMD signals as a function of time from 1995 to 2011 for PMDs 1, 2, and 3 (from top to bottom). Reference measurement is from 3rd July 1995.

factors such as moon availability and phase, non-uniformity of the moon surface, polarization, and partial slit filling. Orbit requirements were so strict that measurements are only possible for a very limited number of orbits per year, with the moon phase always being ~0.6 between full moon and the last quarter. After 2003 no more moon measurements are available. Thus, GDP-L1 does not attempt to generate calibrated radiances for the moon and a long-term analysis of moon observations has not

been performed. Furthermore, for an accurate monitoring of instrument degradation using lunar measurements a more precise characterization of the reflective and scattering properties of the moon would be necessary. Early investigations by Dobber (1997) using the first 18 months of GOME's lifetime confirmed the assumption that the scan mirror (instead of the diffuser) is primarily subject to degradation.

### 3.2 PMD measurements and Q_factors

The relative change of the solar PMD measurements as a function of time with respect to a reference measurement from 3rd July 1995 are shown in Fig. 4 for all three PMDs. Note that the measurements were normalized to 1 astronomical unit (AU) in order to eliminate seasonality. As for the SMR spectra the degradation for the PMDs is strongest for PMD 1 which corresponds to channel 2. The signal decreases to about 40% of the original value. The temporal evolution for PMDs 2 and 3 is similar to the behavior of the signals in channels 3 and 4, respectively.

PMD Q_factors are self-calibration constants which ensure that the calculated fractional polarization $p$ of the Sun is unpolarized with $p = 0.5$. They are defined as the relative difference between the measured solar signal of $PMD_i$, with $i = 1, 2, 3$, and the expected PMD signal calculated from the key data and the corresponding channel signals, when unpolarized input is



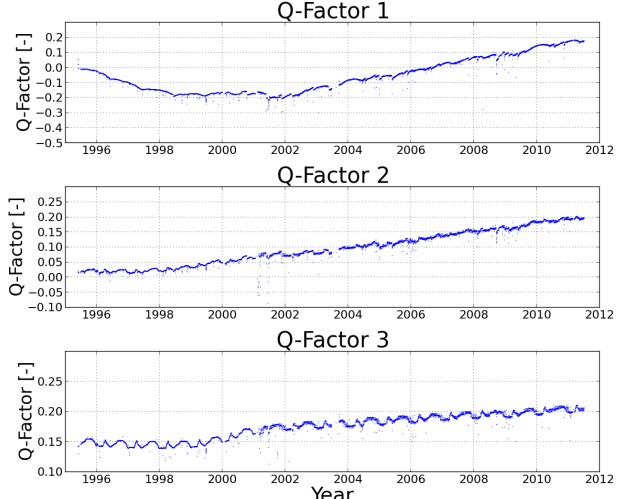

**Figure 5.** Q_factors 1, 2, and 3 (from top to bottom) as a function of time from 1995 to 2011.

assumed:

$$Q\_factor_i = (PMD_i - \sum_j X_j \times channel_j)/PMD_i, \tag{4}$$

where $channel_j$ is the channel signal of pixel $j$ and $X_j$ is the ratio of the PMD signal to the channel signal for a monochromatic input signal as obtained from on-ground calibration measurements. Q_factors thus involve the differential degradation between
PMD signals and the channel signals since the time of on-ground calibration.

Figure 5 shows the Q_factors for PMD 1, 2, and 3 (from top to bottom) as a function of time from 1995 to 2011. In principle, the behavior of the Q_factors as detected in the previous study (Coldewey-Egbers et al., 2008) continued. For the first Q_factor a decrease until 2001 is observed. From 2002 to 2011 Q_factor 1 steadily increased. That means that in the first period the degradation of the PMD signal was stronger than the degradation of the signal in channel 2, whereas in the second period the
channel signal decreased faster then the PMD signal. Q_factor 2 increased slowly from the beginning of the measurements – indicating that the PMD signal degraded less than the average signal in channel 3 – and reached nearly the same value as Q_factor 1 at the end of the mission. For Q_factor 3 note that it is nonzero already at the beginning of the measurements. This is related to stray light (wavelength > 790 nm) which affected in particular PMD 3, whereas PMD 1 had a negligible stray light effect. Q_factor 3 remained more or less stable until 1999 followed by a slow increase until 2011. Outliers are due to GOME
operation anomalies such as cooler switch-offs, instrument or satellite switch-offs, on-board anomalies, or special operations (see also: GOME Mission Operations Overview).





### 3.3 Reflectance degradation

In Sect. 3.1 the approach used to correct for instrument degradation was described. This 'soft' correction is applied to both irradiance and radiance spectra in GDP-L1 thereby assuming that both spectra degrade in the same way. The ratio

$$R = \frac{\pi I}{\mu_0 E}, \tag{5}$$

where $I$ is the Top-Of-Atmosphere (TOA) radiance reflected and scattered by the Earth's atmosphere, $E$ is the solar irradiance, and $\mu_0$ is the cosine of the solar zenith angle, defines the reflectance $R$, which is used by many algorithms to retrieve the amount of atmospheric constituents. From Eq. 5 it is clear that the reflectance remains unchanged in the level 0-to-1 processing since the applied degradation correction cancels out. However, the light paths for radiance and irradiance measurements are different and the degradation of the scan mirror indicates a strong dependence on the incidence angle (Snel, 2001). This leads

to a substantial differential degradation of radiance and irradiance spectra (Tanzi et al., 2001; van der A et al., 2002) and, thus, to degradation in the reflectance, which may affect for example ozone profile retrievals (van der A et al., 2002; Liu et al., 2007) or the determination of total ozone columns using a direct fitting approach (Lerot et al., 2014), because these algorithms are sensitive to absolutely calibrated reflectances. Correction approaches for the reflectance degradation have been developed in the past which rely on, e.g. the comparison of experimental and simulated data (van der A et al., 2002; Cai et al., 2012),

the comparison of satellite reflectance spectra with ground-based reference spectra (Lerot et al., 2014), or the comparison of global average reflectance with respect to global average reflectance from the beginning of the mission (Liu et al., 2007; Tilstra et al., 2012). For the latter approach the underlying assumption is that the global average reflectance does not change in time. For irradiance degradation correction (see Sect. 3.1) this assumption can be regarded fulfilled but the earthshine radiance and, thus, the reflectance on the other hand depend strongly on highly variable atmospheric conditions such as clouds, trace gases,

aerosols, or surface albedo and on the viewing angle. Therefore, retrievals using this correction may be inadequate for trend studies (Liu et al., 2007). On the other hand, Garane et al. (2018) have shown that using the latest version of the direct fitting approach GODFIT for ozone retrieval (GODFIT version 4) GOME (as well as OMI) performs in an extremely stable way and does not require any spectral soft calibration procedure.

In the framework of GOME-Evolution we analyzed the long-term behavior of the GOME reflectance using measurements

over so-called Pseudo Invariant Calibration Sites (PICSs) which have been identified and characterized by the Committee on Earth Observation Satellites (CEOS) to be suitable to detect the radiometric stability of satellite sensors (CEOS Pseudo Invariant Calibration Sites). The advantages of these sites are the spatial uniformity and homogeneity, their stable spectral characteristics over time, and generally high reflectance to enhance the signal-to-noise ratio. At the moment there are six CEOS reference PICSs all located in the Saharan desert: Libya-1, and -4, Mauritania-1 and -2, and Algeria-3 and -5, respectively. They

are usually made up of sand dunes with climatologically low aerosol loading, little rainfall, and practically no vegetation or human impact. More details on the PICSs can be found in Helder et al. (2010). In the past these sites have been widely used in post launch calibration and validation of satellite sensors (e.g., Smith and Cox, 2013; Mishra et al., 2014; Sun et al., 2014; Uprety and Cao, 2015).



For our study we selected four reference sites: Libya-1 (24.42°N, 13.35°E) and -4 (28.55°N, 23.39°E) as well as Algeria-3 (30.32°N, 7.66°E) and -5 (31.02°N, 2.23°E). The geolocation in the parentheses denotes the center latitudes and longitudes. Fortunately, in this area the impact of the ERS-2 tape recorder failure in June 2003 is quite small, so that the time series are almost complete with only a short gap in 2003. We limit our analysis to two single wavelengths in the UV part of the spectrum

(325 and 335 nm, respectively) which mark the lower and upper limit of the fitting window for total ozone retrieval (Loyola et al., 2011; Lerot et al., 2014). All GOME ground pixels with cloud fraction less than 0.2 were extracted that fall into a square area of ±1.5° in latitude and longitude around the center geolocation of the reference site. About 3000 pixels were found for each reference site which fulfill these criteria. In general, the top-of-atmosphere reflectance of a scene measured by the satellite sensor depends on the viewing geometry because of the anisotropy of the surface reflectance. The reflectance is higher for

west pixels, when the sun and the satellite are on the same side of the scene (backward scattering viewing geometry), than for east pixels in forward scattering viewing geometry (Zoogman et al., 2016; Lorente et al., 2018). The anisotropy depends on wavelength and on the surface properties. In case of the Saharan PICSs scenes and for wavelengths of 325 and 335 nm the difference in reflectance between west and east pixels is about 25%.

Figure 6 shows the reflectance normalized to the reflectance at the beginning of the mission, segregated by pixel type (east,

nadir, and west which mark the three forward scans) for both wavelengths (solid curves denote 325 nm and dashed curves denote 335 nm) as a function of time for the entire mission for PICS Libya-4. The data gap from mid 2003 to early 2004 is due to the tape recorder failure. Fluctuations in reflectance related to the seasonal variation and to short-term variation of atmospheric conditions were smoothed out. Until late 1999 the curves show only little degradation. Afterward they start to increase, reach a maximum in 2001, and then decrease again to values below 1 in early 2003. From 2004 to 2011 the curves

steadily increase except for the reflectance degradation for west pixels at 325 nm (solid cyan curve) which show a slight decrease at the very end of the mission. The degradation depends strongly on the pixel type, i.e. the line-of-sight. Until 2003 west pixels are much less affected than nadir and east pixels, and also after 2003 the behavior of the reflectance from west pixels is slightly different compared to nadir and east pixels. For the other three PICSs Libya-1, Algeria-3, and Algeria-5 we found very similar results (without figure) with negligible differences compared to Libya-4. In addition, we analyzed the reflectance

for cloud free pixels over the Mediterranean Sea between Greece and Egypt, which shows the same temporal evolution as the reflectance over the Saharan desert (without figure) although the surface albedo is much lower there. As mentioned earlier, in principle this analysis could be used for correcting the reflectance degradation. However, the underlying requirement that the reflectance remains stable over a long time period might not be fulfilled in every case.

## 4   GOME in-flight calibration parameters

In this section we present the analysis of the GOME calibration parameters obtained from measurements using on-board calibration sources and applied during the level 0-to-1 processing as described in Sect.2.2. For a detailed description of the individual calibration algorithms related to the parameters we refer to the GOME Algorithm Theoretical Basis Document





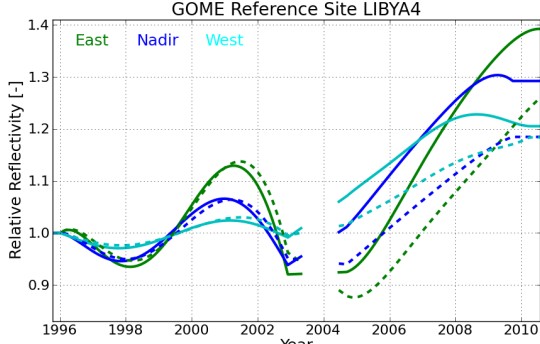

**Figure 6.** Relative reflectivity as a function of time for cloud-free GOME measurements at 325 nm (solid) and at 335 nm (dashed) for Libya-4 reference site. Colors denote the different GOME pixel types: east (green), nadir (blue), and west (cyan) ground pixels. The data gap between mid 2003 and early 2004 is due to the ERS-2 on-board tape recorder failure in June 2003.

(Slijkhuis and Aberle, 2016). Monitoring the individual parameters was performed with special emphasis on the analysis of the long-term stability.

### 4.1 Overview

In the framework of the GOME-Evolution project the complete set of in-flight calibration data has been revisited and re-analyzed in order to draw conclusions on the long-term stability of the GOME sensor and to optimize the GDP-L1 usage of the in-flight calibration for the entire mission. The database contains spectral lamp measurements for the wavelength calibration (see Sect. 4.2), dark current measurements for all integration time patterns (see Sect. 4.3), LED measurements for the pixel-to-pixel gain correction (see Sect. 4.4), as well as the Sun mean reference spectra, and moon and PMD measurements. After the ERS-2 tape recorder failure in June 2003 the number of available calibration data is significantly reduced since only data within accessibility of an ERS-2 receiving station were transmitted to ground. In particular, no more moon measurements are available after 2003.

### 4.2 Spectral calibration

The objective of the spectral calibration is to assign a certain wavelength to each individual GOME detector pixel. Therefore, the instrument houses a platinum-chromium-neon hollow cathode emission lamp (Murray, 1994). This lamp provides a sufficient number of atomic emission lines of these three elements with well-known spectral positions which allow the wavelength allocation. At first, spectral calibration parameters are calculated by the determination of the pixel number center of the spectral lines and the subsequent fitting of a polynomial through these pixel-wavelength pairs. The second step is the application of the calibration parameters from the previous step to the measurements.





Several lamp spectra were measured (i) over the orbit approximately once per month, during the calibration timeline which was run for five orbits, and (ii) every day just before and after the Sun calibration. The latter measurements are available until April 1998. Since September 2001 the calibration lamp was used only during the five orbits of the monthly calibration due to numerous lamp failures since the voltage has not reached its nominal value (see also GOME Mission Operations Overview).

For the spectral calibration a total of 68 candidate emission lines within GOME's spectral range from 240 to 790 nm has been selected from the reference lamp atlas (Murray, 1994) and is stored in the calibration key data base. The lamp measurements of the individual lines can be regarded as statistical distributions from which the moments can be calculated. They contain characteristic information about the spectral lines that are needed to select those lines suitable for an accurate calibration. The aforementioned moments are the mean value, i.e. the pixel number center of the maximum intensity, as well as the variance,

the standard deviation $\sigma$, and the skewness $skew$. The full width half maximum (FWHM) is computed from the standard deviation. To be selected, the moments of a spectral line must meet the following statistical criteria: (i) the signal of the center pixel shall not be below a certain minimum, i.e. well above the noise level, (ii) the FWHM shall not be below a certain value in order to fulfill the Nyquist criteria for the digital recording of analogue signals, and (iii) the skewness shall not be larger than a certain value, i.e. the line must be roughly symmetric. Reasonable thresholds for the criteria have been determined during the

pre-flight measurements and the commissioning phase. Current values are 50 BU/s for channel 1 and 300 BU/s for channels 2–4 for the first criterion, $\sigma \geq 0.6$, FWHM $\geq 1.5$ pixel, and $skew \leq 0.6$ for the second and third criterion, respectively.

As mentioned before, the calibration parameters are obtained by fitting a polynomial through the pixel-wavelength pairs. In channels 1 and 2 third-order polynomials are used, whereas in channels 3 and 4 fourth-order polynomials are used, respectively. At least seven spectral lines per channel are needed for the fit which is performed using the singular value decomposition

algorithm (Press et al., 1992).

The statistical parameters of each individual emission line were analyzed in terms of both long- and short-term stability. Regarding short-term variability few lines were found whose moments show jumps between two values leading to jumps in the fitted polynomial coefficients. Other lines fulfill the aforementioned criteria only in very few cases which also results in jumps in the fitted coefficients. This analysis has lead to a revised spectral line list (by excluding the identified unstable lines)

that improved the stability of the spectral calibration for the complete mission. Figure 7 shows the wavelength changes of selected lamp lines (two per channel) as a function of time. Depicted is the difference (in nm) with respect to the wavelength at the beginning of the mission. The stability of the wavelengths is excellent until 2004. Toward the end of the mission the variability increases slightly, in particular in channels 1 and 3. The standard deviation of the wavelength changes is 0.0015 nm in channel 1, 0.0025 nm in channel 3, and less than 0.001 nm in channels 2 and 4, respectively. These values are comparable to

the analysis by van Geffen (2004) who used a different wavelength calibration approach (van Geffen and van Oss, 2003). They found temporal variations of the wavelength calibration from 0.0015 to 0.0034 nm for nine narrow spectral bands.

One of the key elements in the optical system of GOME is a quartz predisperser prism. The wavelength calibration is sensitive to the dispersion of this prism, whose refractive index varies with temperature. Thus, the calibration parameters from the lamp measurements are stored in the database as a function of this temperature. In the operational processing the most

recent calibration parameters are then selected from the database according to the predisperser temperatures encountered in





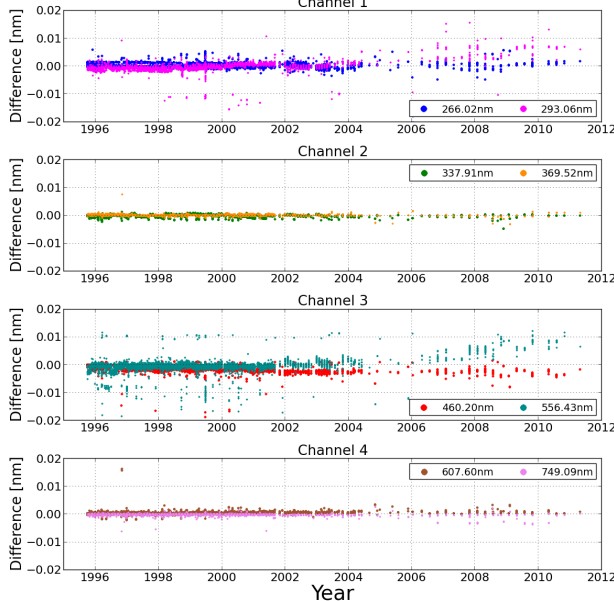

**Figure 7.** Change in wavelength (with respect to the beginning of the mission) as a function of time from 1995 to 2011 for two selected lamp lines per channel. From top to bottom: channel 1, channel 2, channel 3, and channel 4.

the actual orbit. Each individual GOME spectrum is, thus, implicitly corrected for temperature variations that are caused by seasonal variations, the position in the orbit, and by the rate of degradation of thermally sensitive optical elements.

Figure 8 shows the time series of the predisperser temperature from 1995 to 2011 (blue dots). An increase of $\sim$4 K within the instrument's lifetime is found which is due to degradation of the thermal system. Furthermore, the curve exhibits a seasonal
5     cycle with maximum values in December/January when the Sun-Earth distance is at a minimum. Outliers are caused by instrument and cooler switch-offs. Magenta dots denote the increase of the predisperser temperature along an orbit in Kelvin hour$^{-1}$ for the years 1995 to 2003. The increase along an orbit is due to warming of the satellite by the Sun and because light passes through the instrument. This analysis relies on the average of 60-70 days per year; for each day the temperature measured along the first orbit, which is always located between 120-160°E, was investigated. The errorbars are a measure of the
10    intra-annual variability. We did not analyse the dependence of the predisperser temperature itself on longitude as in van Geffen (2004, his Fig. 3) who found a maximum over the Atlantic and a minimum over the Pacific. However, they also stated that the temperature increase along the orbits does not show a dependence on longitude. During the first eight years of the mission the temperature raise along the illuminated part of one orbit increased from about 0.7 K/h to about 0.9 K/h. Unfortunately, analysis of later years is not possible due to the ERS-2 tape recorder failure and incomplete orbits.





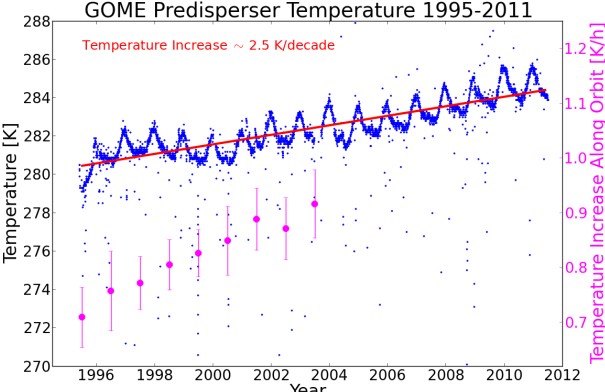

**Figure 8.** Temperature measured at the pre-disperser prism as a function of time from 1995 to 2011 (blue) and linear fit (red). The temperature increase is about $2.5\,\text{K}\,\text{decade}^{-1}$. Magenta dots denote the increase of the pre-disperser temperature along one orbit in Kelvin hour$^{-1}$ (right y-axis) for the years 1995 to 2003. Analysis of later years is not possible due to the tape recorder failure (incomplete orbits).

### 4.3 Dark signal correction

The detectors integrated in GOME are random access linear photo-diode arrays. One of the characteristics of these devices is a certain amount of dark current due to thermal leakage. It is expected that this current will depend on the orbital position of the satellite and also the time into the mission. Therefore it is necessary to continuously monitor the dark current and the associated noise which is done by means of periodically taken dark-side measurements (Sect. 4.3.1). In this case the scan mirror points toward the GOME interior. The PMD detectors are non-integrating devices and, therefore, do not have a leakage current. Nevertheless, those detectors must be corrected for their zero offsets and the noise must be monitored (see Sect. 4.3.2).

#### 4.3.1 Dark current and dark current noise

The complete dark signal comprises two parts: (i) a constant value of ∼140-150 binary units (BU) which is called the fixed pattern readout noise (FPRN) and (ii) the time-dependent leakage current itself which is about ∼2 BU/s. This value is quite small because of the low temperature (-38 °C) of the detector arrays. The dark signal measurements have to be taken with the same integration time patterns as those used for scanning and other calibration measurements since it was found that a certain amount of cross talk is present which depends on the integration time. However, the detector temperature is not taken into account for GOME as it is the case for the dark signal correction of the GOME-2 instrument (Munro et al., 2016).

The dark signal correction is the subtraction of a mean dark signal spectrum from the measured signal $S_i^{meas,k}$:

$$S_i = S_i^{meas,k} - \overline{S_i^{dark,k}}, \tag{6}$$

where $i = 1, ..., 1024$ detector pixels. The integration time pattern $k$ describes the number of clock pulses, where one pulse takes 93.75 ms, e.g., a time pattern of 640 is equivalent to 60 s.





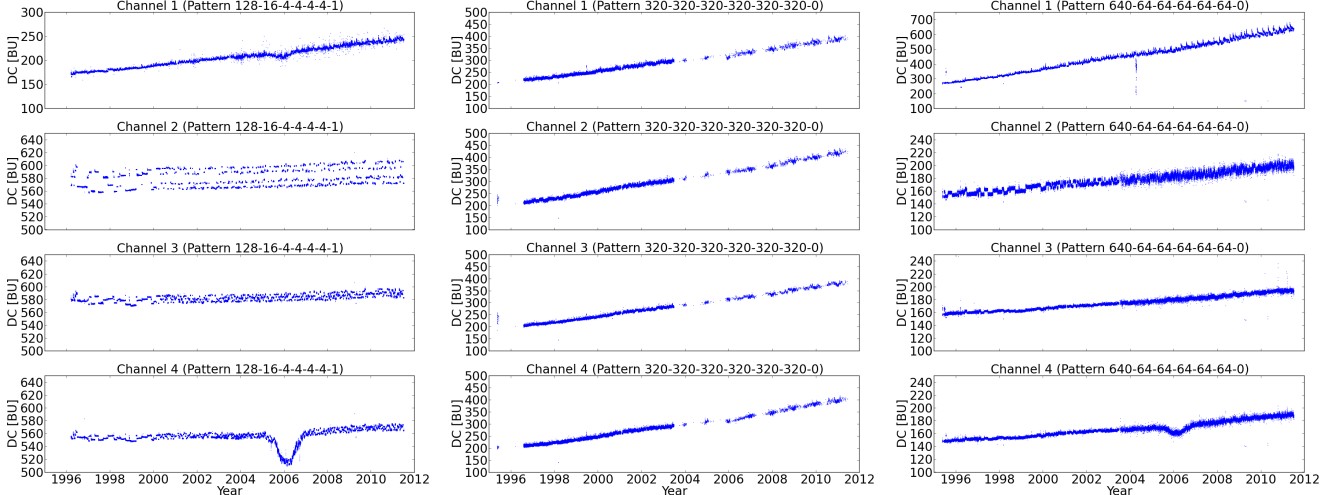

**Figure 9.** Leakage current in binary units (BU) as a function of time from 1995 to 2011 for three integration time patterns (see text for more explanation): normal scanning mode (left), LED measurement mode (middle), and polar view mode (right). From top to bottom: band 1a, band 2b, band 3, and band 4.

The mean dark signal for $n = 10$ consecutive measurements is defined as

$$\overline{S_i^{dark,k}} = \frac{1}{n} \sum_{j=1}^{n} \left( S_i^{dark,k} \right)_j. \tag{7}$$

Figure 9 shows the dark signal as a function of time for the three most representative integration time patterns: (i) the normal scanning orbits with 12 s integration time for band 1a and 1.5 s for the other bands (with co-adding applied), (ii) the LED measurements for the pixel-to-pixel gain correction (see next Section) with 30 s integration time for all bands, and (iii) the polar view mode with 60 s integration time for band 1a and 6 s for the other bands. Figure 9 shows the dark signal for bands 1a, 2b, 3, and 4 (from top to bottom) and for time patterns (i) to (iii) from left to right. All panels denote a significant increase in the leakage current over time.

Note that the dark signal in bands 2b, 3, and 4 for the normal scanning orbits (Fig. 9, left panel) is much higher due to co-adding of four measurement sequences. At present there is no explanation for the behavior of the signal from 2005 to 2007. It is most obvious in channel 4 for the normal scanning mode (bottom left panel) and for the polar view mode (bottom right panel). The signal decreased significantly in 2005 (by 40 BU for the normal scanning mode), reached a minimum in the beginning of 2006 and increased again, in which the entire development of this anomaly is quite smooth. The jumps in the time series (e.g., seen in channels 2 and 3 for the normal scanning mode and in channel 3 for the polar view mode) are due to instrument or cooler switch-offs or instrument anomalies.

The noted increase in the dark signal is an increase in the second part of the leakage current, i.e. the time-dependent part. Figure 10 shows the leakage current as a function of the integration time for four different years 1997, 2002, 2007, and 2011. Different symbols and linestyles denote the individual bands 1a, 2b, 3, and 4. The y-intercept represents the FPRN which is





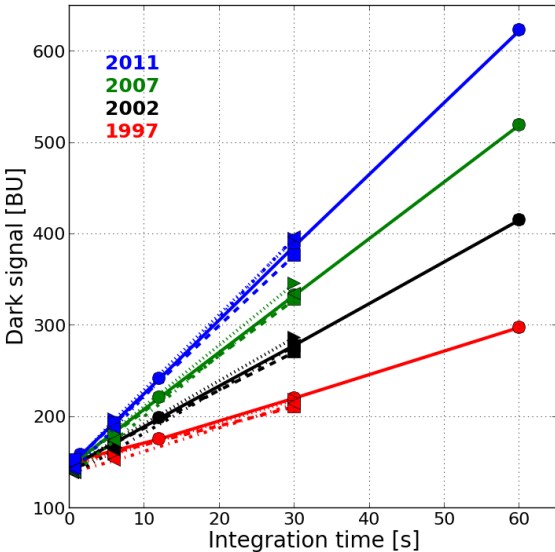

**Figure 10.** Dark signal in binary units (BU) as a function of integration time for January 1997 (red), 2002 (black), 2007 (green), and 2011 (blue), respectively. Different symbols and linestyles denote channels 1a (solid), 2b (dotted), 3 (dashed), and 4 (dash-dotted).

about 140-150 BU and remains constant over the entire time period. The slope denotes the time-dependent leakage current which is quite similar for all channels ($\sim$2 BU/s) and which increases over time. The increase is also almost identical for all channels and amounts to about 4 BU/s per decade ($\sim$6.5 BU/s from 1995 to 2011). This is comparable to earlier work by Dehn (2003) and our predecessor study (Coldewey-Egbers et al., 2008) as well as to Munro et al. (2016) who analyzed the dark

signal for the GOME-2 instrument on-board the MetOp series of satellites using the same type of detectors. For OMI, which is a nadir viewing UV-VIS imaging spectrograph using two-dimensional charge-coupled device (CCD) detectors (Levelt et al., 2006; Dobber et al., 2006), a 7-fold dark current increase was found from 2005 to 2015 (Schenkeveld et al., 2017), and for GOMOS/ENVISAT (Global Ozone Monitoring by Occultation of Stars) – using the same CCD detectors as OMI an even higher increase was found (Bertaux et al., 2010). Although the increase in the dark current seems to be significant, there is not

necessarily a negative impact on the quality of the level 1 data products as long as appropriate dark current measurements are available and applied during the level 0-to-1 processing.

We found that it is not only the leakage current itself which changed over time, but also its distribution which considerably widened. Figure 11 shows histograms of the dark signal for spectral band 1a (240–283 nm) for an integration time of 12 s (nominal scanning mode) for every two years from 1997 to 2011. The data correspond to the upper left panel of Fig. 9.

Colored numbers stuck to the individual histograms denote the median values and the FWHM of the distribution. The latter is additionally indicated by the filled rectangles. As seen in Fig. 9 the dark signal significantly increased over time from $\sim$176 BU to $\sim$242 BU. Furthermore a noticeable, almost three-fold, broadening of the distribution was found. FWHM increased from

(c) Author(s) 2018. CC BY 4.0 License.



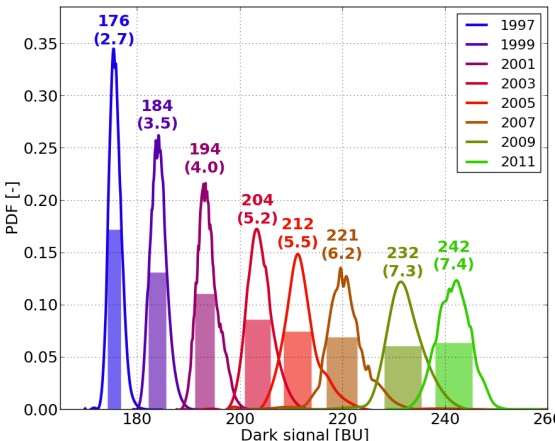

**Figure 11.** Histograms of the dark signal in spectral band 1a for an integration time of 12 s for every two years from 1997 to 2011. Colored numbers denote the median value in binary units (BU) and the FWHM (in parenthesis) of the distribution. The latter is additionally indicated by the colored rectangles.

2.7 BU to 7.4 BU. A widening of the dark current distribution was also noticed for OMI (Schenkeveld et al., 2017) and GOMOS (Bertaux et al., 2010).

The noise on the signals of the detector pixel readouts is expected to be constant over all individual pixels. For each detector pixel standard deviation from all leakage measurements from one orbit with the same integration time is computed. The noise is then the average of all standard deviations. The annual mean noise level is shown in Figure 12 (blue curves, left y-axis) as a function of time for three different integration time patterns (scanning, moon and LED). The error bars denote the standard deviation for the annual mean. The lowest noise level (∼2 BU) is found for LED dark signal calibration measurements which have the longest integration time (30 s), whereas the noise level for scanning and moon integration time pattern are quite similar and about 4 BU. The values remain more or less constant until June 2003. Afterward the noise level for LED dark signal calibration measurements slightly increased (dot-dashed blue line), whereas a decrease is found for moon dark signal calibration measurements (dashed blue curve). Red curves (right y-axis) denote the number of available dark signal calibration measurements. In case of LED dark signal calibration measurements the most significant decrease in the number of available measurements is found (dot-dashed red curve).

### 4.3.2 PMD offset and noise

The signals of the PMD detectors as non-integrating devices must be corrected for their zero offsets and the associated noise must be monitored. Figure 13 shows PMD offsets for each PMD as a function of time for the entire mission period. The offset of PMD 1 is about 1320 BU, whereas it is about 510 BU for PMDs 2 and 3. All offsets indicate a very small increase of 0.8% in 16 years of the mission. The increase is nearly linear for PMDs 2 and 3, whereas for PMD 1 the increase started in 1999;





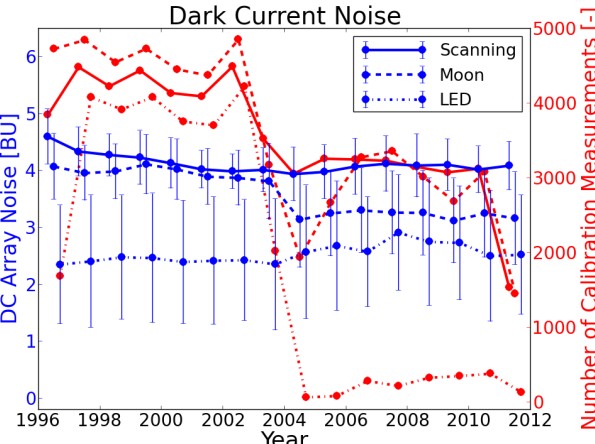

**Figure 12.** Annual mean dark current noise (blue curves, left y-axis) in binary units (BU) for three integration time patterns: scanning (solid curve), moon (dashed curve), and LED (dot-dashed curve). Red curves (right y-axis) denote the number of available calibration measurements for the three individual time patterns.

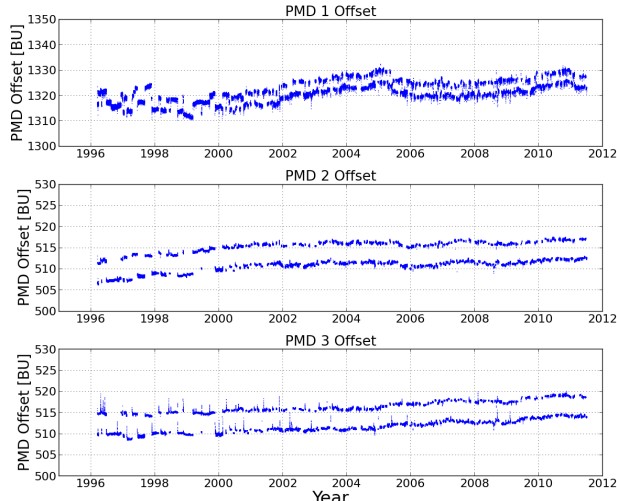

**Figure 13.** PMD offset in binary units (BU) as a function of time for PMD 1, 2, and 3 (from top to bottom).

the PMD 1 offset reached a maximum at the end of 2004, decreased in 2005 and increased again thereafter. For all PMDs the offsets seem to have two states, and jumps between the two states are due to cooler and instrument switch-offs as well as instrument anomalies.

Figure 14 shows the annual mean PMD noise as a function of time. The PMD noise is defined as the mean value of the standard deviations which are calculated for each PMD over all 16 individual PMD measurements. It is about 0.5-1.5 BU.



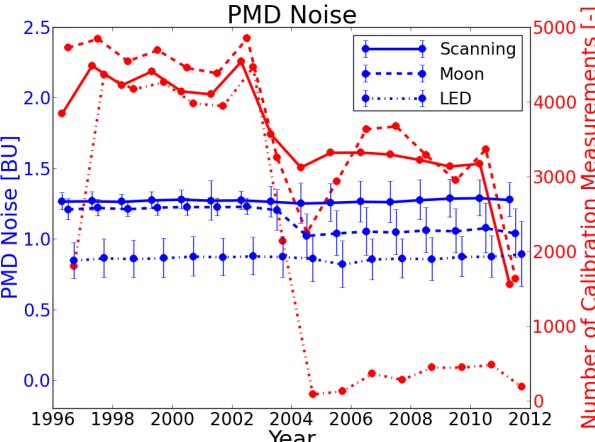

**Figure 14.** Annual mean PMD noise (blue curves, left y-axis) in binary units (BU) for three integration time patterns: scanning (solid curve), moon (dashed curve), and LED (dot-dashed curve). Red curves (right y-axis) denote the number of available calibration measurements for the three individual time patterns.

The previous study (Coldewey-Egbers et al., 2008, their Fig. 9d) has shown the impact of the South Atlantic Anomaly on the noise level which increases significantly when measurements from this area are taken into account. In the new GDP-L1 version these calibration measurements are discarded (see previous section). In general the noise level remains stable over the entire period although – as a consequence of the tape recorder failure in June 2003 – a slight change in the noise level was found, in particular for the moon dark signal calibration measurements.

### 4.3.3   Impact of South Atlantic Anomaly

The South Atlantic Anomaly (SAA) is an area of enhanced flux of energetic particles due to a dip in the Earth's inner Van Allen radiation belt. In this region low Earth orbit spacecrafts are exposed to higher-than-normal radiation levels and may suffer from damage (Heirtzler, 2002; Casadio and Arino, 2011). High energy protons impact the detectors of GOME, i.e. the background signal is higher than the normal dark signal, the noise is enhanced, and the measured spectra are also prone to intensity spikes caused by cosmic particles.

For this reason all calibration measurements in the SAA are discarded. The algorithm to identify the SAA uses the signal from PMD 1 since it has been found that the noise level on PMD 1 is a reliable indicator of the enhanced particle bombardment in the SAA region. Figure 15 shows a map of the GOME long-term mean PMD 1 noise derived from the first five years of the mission. The impact of the SAA clearly appears in terms of significantly enhanced PMD 1 noise in an oval-shaped region centered at the east coast of Brazil. The SAA spans from 50°S–0°in latitude and from 90°W–30°E in longitude. During the level 0-to-1 processing PMD measurements are grouped and for each group a noise value w.r.t. the median value is calculated. If the noise value exceeds a certain threshold all calibration measurements from the group are discarded. This includes also





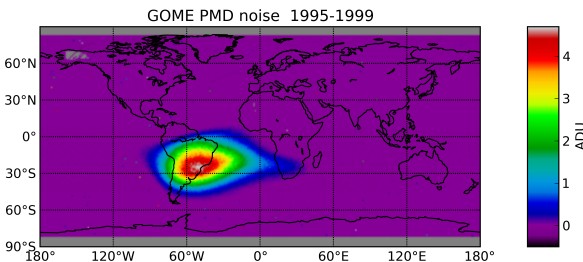

**Figure 15.** Map of the GOME long-term mean PMD 1 noise in A/D[-converter] Units (= binary units) derived from the first five years of the mission. Enhanced noise levels indicate measurements affected by the South Atlantic Anomaly.

the lamp measurements for the spectral calibration and the LED measurements for the pixel-to-pixel gain correction (see next Section). The new algorithm defines an 'inside SAA' and an 'outside SAA' region for dark signal values in the calibration database.

## 4.4 Pixel-to-pixel gain correction

The pixel-to-pixel (PPG) variability in quantum efficiency of each diode detector array is characterized and corrected using internal LEDs. Each channel has a monochromatic red LED located between the channel optics and the detector window (see Fig. 1), i.e. the detectors are illuminated directly without any dispersing element in between that may suffer from degradation effects. The monitored detector signal corresponds to a superposition of a smoothly varying signal caused by the LED characteristics and a small-scale structure due to the slightly different sensitivity of each pixel. The determination of the correction

spectra for each of the four channels is based on a mean value of several consecutive LED measurements and a smoothed curve through this average using a triangle filtering window:

$$c_i = \frac{S_i^{smooth}}{\overline{S_i^{LED}}}, \tag{8}$$

where $c_i$ is the correction factor of detector pixel $i$, $\overline{S_i^{LED}}$ is the mean value of several consecutive LED measurements, and $S_i^{smooth}$ is the smoothed curve through this averaged measurements. The latter is calculated by means of

$$S_i^{smooth} = \frac{\sum_{k=-n}^{n} \frac{n-|k|}{n} \times \overline{S_{i+k}^{LED}}}{\sum_{k=-n}^{n} \frac{n-|k|}{n}} \tag{9}$$





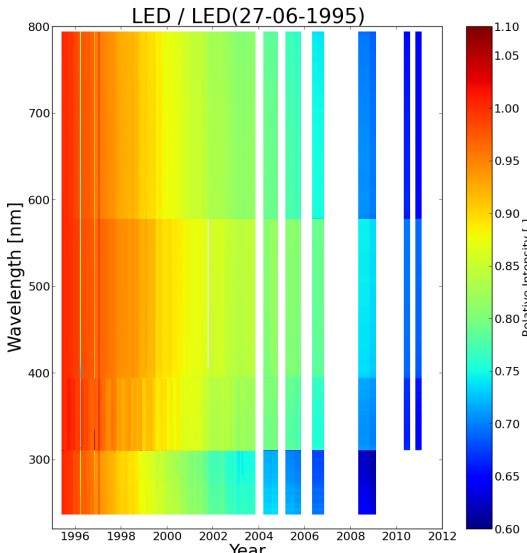

**Figure 16.** Relative intensity of LED spectra as a function of time (x-axis) and wavelength (y-axis) with respect to a reference spectrum from 27 June 1995.

using a triangle filtering window of width $n = 5$. The application of the PPG correction is then simply

$$S_i^{corr} = S_i c_i, \tag{10}$$

where $S_i$ is the measured signal value of detector pixel $i$, and $S_i^{corr}$ is the corrected value.

   Typically, the LED spectra were obtained in monthly intervals until 2003. From 2003 onward LED measurements are limited
5  to two or three sequences per year. The absolute radiance correction due to the pixel-to-pixel variability is very small (∼0.02%).
   However, it may not be negligible in wavelength regions used for the retrieval of weak absorbers such as bromine oxide. Figure
   16 shows the relative intensity of the LED spectra as a function of time and wavelength with respect to a reference spectrum
   from the beginning of the measurements (27 June 1995). The nearly linear decrease which was already detected in the previous
   study (Coldewey-Egbers et al., 2008) continued until the end of the mission in 2011 and is due to the degradation of the LEDs'
10 brightnesses themselves. The output decreased to ∼60%. It is almost homogeneous over the complete wavelength range of
   each channel. The steepest decrease is found in channel 1.

   In addition we analyzed the distribution of the PPG correction factors as a function of time. Figure 17 shows box-whisker
   plots of the distribution for each channel and as a function of time. We show one distribution per year. In channel 1 the amplitude
   of the PPG correction spectrum is slightly larger than for the other channels. Nevertheless, the distribution of the correction
15 spectrum remains roughly stable over the entire period, whereas for channel 2 a significant broadening of the distribution is
   found. The standard deviation increased by a factor of ∼2.5 in this channel, which indicates that the variability in sensitivity




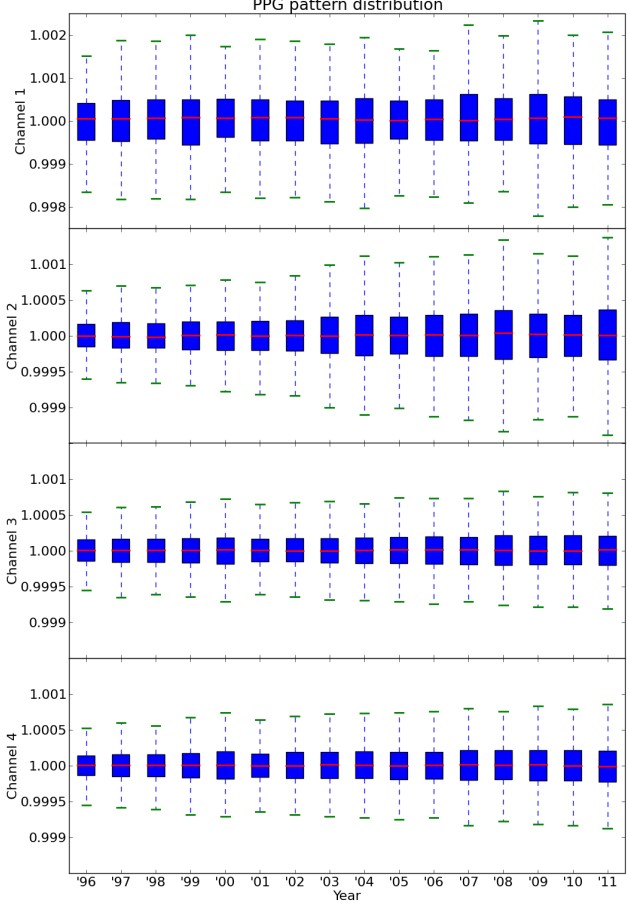

**Figure 17.** Box-Whisker plots of the distribution of the PPG correction pattern as a function of time (one selected distribution per year) for channels 1, 2, 3, and 4 (from top to bottom). Note the slightly different y-axis range for channel 1. Red horizontal lines denote the median, the blue boxes denote the lower (25%) and the upper (75%) quartile, and the green caps denote the minimum and maximum values (except the outliers).

between the individual pixels increased significantly. For channels 3 and 4 a broadening of the distribution of ∼40% was found. For all channels we noticed that the number of outliers did not increase over the years (not shown in this plot) which indicates that the detector as a whole is affected and that the increase is not just due to a few strongly battered pixels.

## 5 Summary and conclusions

5    The Global Ozone Monitoring Experiment, launched in April 1995 on-board the second European Remote Sensing satellite, provided measurements of atmospheric trace constituents such as $O_3$, $NO_2$, $SO_2$, HCHO, BrO, $H_2O$ as well as aerosol and cloud parameters on a global scale for more than 16 years, before it was decommissioned in July 2011. The existing data





archive of GOME can be considered as the European reference for follow-up atmospheric composition sensors like SCIA-MACHY, OMI, GOME-2, and the Copernicus Sentinel missions S5P/S4/S5, and preservation as well as further improvement and exploitation of this unique data set are highly recommended.

Within the framework of the ESA's GOME-Evolution project a homogenized level 1 data product for the complete mission was generated for the first time, based on the new GDP-L1 Version 5.1, that contains fully calibrated radiances, irradiances, geolocation information, and selected calibration parameters. In addition, cloud parameters retrieved with the well-established OCRA and ROCINN algorithms have been integrated in the new product. The format and structure of the GOME L1 NetCDF-4 files is similar to other state-of-the-art EO products like S5P. Furthermore, a detailed investigation of the long-term performance of the GOME instrument in terms of monitoring the various in-flight calibration parameters was carried out. This should ensure the high quality of the GOME (ir)radiance measurements that is needed to retrieve atmospheric geophysical products with highest accuracy.

By means of the daily solar irradiance measurements the degradation was monitored. Degradation can be explained in terms of deposits on the GOME scan mirror. Below 300 nm intensity decreased by 80-95% which implies a significant deterioration of the signal-to-noise ratio and which may have a severe impact on the challenging retrieval of atmospheric parameters such as ozone profiles. The decrease in channel 2 is 40-80%. In channel 3 the decrease (10-40%) started in 2001 when the measurements were additionally affected by an ERS-2 pointing problem. Throughput changes in channel 4 are relatively small. A correction algorithm has been developed and improved which relies on the intensity measured in the early part of the mission and which comprises a wavelength- and a time-dependent part. The degradation in reflectance, i.e. the differential degradation between solar irradiance and earth radiance measurements has been monitored using cloud-free pixels over the Saharan desert.

For the spectral calibration special attention was paid to the identification of lamp lines that remain stable (with respect to the statistical moments) over the whole mission. This has resulted in an updated spectral line list that improved the temporal stability of the wavelength assignment. For the leakage current an increase of 4 BU/s per decade was found. In general, this has not necessarily a negative impact on measurement quality as long as appropriate dark signal measurements from the same orbit are available and applied. The existing dark signal correction has been further improved by differentiating between measurements from outside and inside the SAA. The output of the LEDs that are used to monitor the pixel-to-pixel sensitivity decreased to about 60% of the early-mission values. For channel 2 a significant broadening of the PPG distribution was observed.

*Data availability.* The new GOME L1 products can be accessed via https://earth.esa.int/FastRegistration/ER**_GOM_SPC_1P-A.

## Appendix A: NetCDF structure of new level 1 files

The GOME level 1 product filename is constructed as follows:

<MMM>_<CCCC>_<TTTTTTTTTT>_<instance ID>.nc,





where <MMM> is the mission ID, <CCCC> is the file class, and <TTTTTTTTTT> (= <FFFF><DDDDDD>) is a mission specific file type. <FFFF> is the file category and <DDDDDD> is a product semantic descriptor. <instance ID> consists of start time, end time, orbit number, packet version, processor version, and processing time. The packet version is a version number which is specific for the combination of processor version, input data (for example calibration data) version and configuration version. For GOME Level 1 products, the mission ID is ER2 for ERS-2. The file class can be TEST or RPRO for test data or reprocessing. The file type field contains an instrument identifier (GOM) as file category and the processing level (L1B_ or L2__ or L0__) as semantic descriptor. The packet version is 2 and the processor version is currently 5.1. We encode these versions into "02_051000" as the versions part of the "instance ID". The file extension is ".nc" that is typically used for netCDF files. All time strings in the filename and product are formatted in ISO 6801 format. Following this scheme, the result would be for example: ER2_TEST_GOM_L1B____20010811T032404_20010811T050712_32981_02_051000_20150311T151024.nc. Product size may vary between 60 and 75MB. Products which are measured after the ERS-2 tape recorder problem in June 2003 are typically smaller because they don't comprise measurements for the entire orbit.

The different dimensions in the GOME level 1 netCDF file are time (=1), which corresponds to one time per orbit, scanline (≈500), which corresponds to one complete scan comprising three forward and one backward scan, ground_pixel (3 or 1), which corresponds to the number of across-track scans, detector channel (=4), which corresponds to the number of detectors, band (=6), which corresponds to the number of spectral windows, and spectral_channel, which corresponds to the total number of detector pixels.

Figure A1 provides an overview of the netCDF structure of the level 1 file. In addition to metadata and instrument related parameters, calibration data and irradiance measurements are available. The radiance measurements themselves are organized in groups for different modes: nadir, static_view, narrow_swath, north- and southpolar_view, sun, or moon. Forward and backward scans are separated in different groups. Inside these groups there are subgroups for bands and PMDs. A band is a part of a channel which can have its own integration time and co-adding factor. Integration times may change during one orbit. All subgroups contain several variables and attributes.

*Competing interests.* The authors declare that they have no conflicts of interest

*Acknowledgements.* This work was performed in the framework of the ESA GOME-Evolution project. Particular thanks to Wolfgang Lengert, ERS-2 Mission Manager, for making this work possible. Thanks to Kai-Uwe Eichmann (IUP-B), Andreas Richter (IUP-B), Mark Weber (IUP-B), Steffen Beirle (MPIC), and Gabriele Brizzi (SERCO) for helpful feedback on the prototype version of the new GOME L1 product.





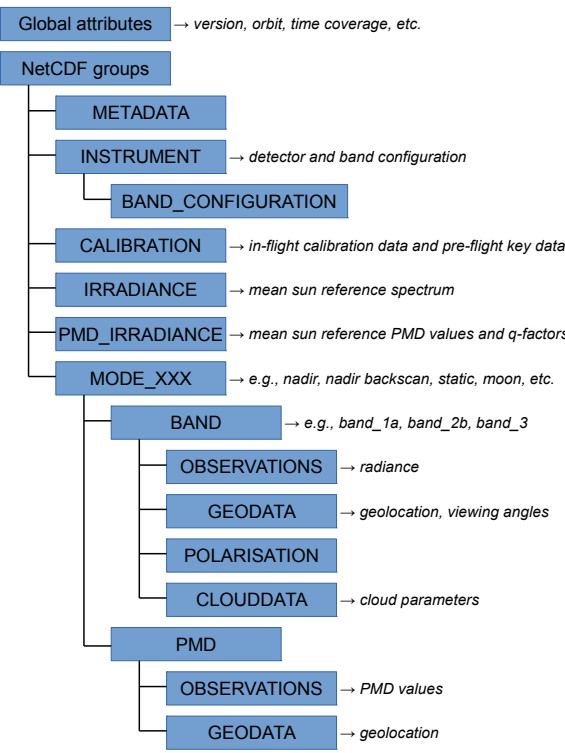

**Figure A1.** NetCDF structure of new GOME level 1 file version 5.1. Measurements are organized in groups for different modes and bands (see text for more explanations).

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
