# Peer review of "The Global Ozone Monitoring Experiment: Review of in-flight performance and new reprocessed 1995–2011 level 1 product"

_Atmospheric Measurement Techniques, 2018_

## Referee Comment (RC1) · Anonymous Referee #1 · 7 Jun 2018

**1. General comments**

The paper is well written and gives a good overview of the re-processing of GOME data together with its in-flight performance during the entire mission. The new level 1 product and new results on the instrument monitoring are definitely worth a long publication.

I find it a very good and practical idea to include a description of the new data format in the appendix, this will give the users a good starting point.

Although the overall quality of the paper content is already high, there are a number of points where the structure and phrasing can be changed to improve the overall readability.

It is not clear, which of the new in-sights and degradation corrections are included in the re-processed data set. An overview (table or graph) of the main level 1 processing steps indicating the changed steps might be an easy remedy for this. Are all the long term monitoring drift and degradation corrections included in the (ir-)radiance data now?

New users of GOME data would be helped by a few brief explanations on the mentioned GOME specifics. Especially if this new data-set is to become a reference, the paper should be as stand-alone as possible.

For the different radiometric steps a few sentences at the beginning of Section 3 would be helpful. There you could explain that the main cause for degradation is the scan mirror, the biggest correction is done based on irradiance monitoring , then the differences between radiance and irradiance degradation are corrected for in the reflectance. Explaining the approach first and then going into the details would improve the readability of this part a lot.

Section 2 of this review lists specific comments and questions about the content and understanding of the paper. These also include the issues summarized in the general comments. Once these points are clarified, I would strongly recommend the paper for publication.

Section 3 contains suggestions where to edit the text to allow for a smoother read, minor spelling and grammar errors and formatting issues.

This review is based on the version amt-2018-118.pdf retrieved on the 4[th] of June from https://doi.org/10.5194/amt-2018-118. The earlier version (amt-2018-118-manuscript-version1.pdf retrieved on the 1st of June) was not considered.

**2. Specific comments**

| # | Page | Line | Section | Comment |
|---|---|---|---|---|
| SC1 | 1 | 21 | 1 | You could also add S5P, S5, S4. |
| SC2 | 3 | 12 | 2.1 | Is the integration time for the forward scan also 1.5 s? This is not clear from the text. |
| SC3 | 4 | 12 | 2.1 | What kind of channel separator? In line 14 a di-chroic is mentioned, is a di-chroic used here too? |
| SC4 | 5 | 5 | 2.2 | Please add which parameters are calibrated in-flight and which are based on on-ground data only. |
| SC5 | 5 | 7-17 | 2.2 | It would be very useful to have a flowchart showing the actual order of the steps, also indicating which steps have been changed/improved with the new version. Maybe even show which step is based on in-flight data. The word "steps" in line 7 suggests that the list shows the order of the steps. Consider writing "the basic calibration algorithms are:" instead. |
| SC6 | 6 | 8/9/10 | 2.2 | Are these calibration constants then fed back immediately into the L1 processor? |
| SC7 | 6 | 9 | 2.2 | The use of the word " comprise" is confusing here. It suggests that the list is complete, that only data from dark measurements, the PtNeCr lamp and the LED is used to directly derive calibration constants during L1 processing. Is this indeed correct? I would also expect that for example transient filtering is performed for all data during L1 processing. |
| SC8 | 6 | 30 | 2.3.1 | What are the other sources? What is "slow" noise? "Slow" with respect to what? |
| SC9 | 6 | 32 | 2.3.1 | "from one typical orbit": Isn't this section about on-ground calibration data? Does this scaling factor ever change? |
| SC10 | 6 | 33 | 2.3.1 | So only band 1a earthshine needs this correction as all other bands and modes have a shorter integration time? |
| SC11 | 7 | 2 | 2.3.1 | Are all Peltier signals involved for all detectors? Or only the Peltier belonging to band 1 a? Please clarify. |
| SC12 | 7 | 6/7 | 2.3.1 | So the additional residual correction is not part of the L1 processor discussed here? Was it not feasible to include this? |
| SC13 | 7 | 12 | 2.3.2 | "In-flight calibration exercise" : do you actually include in-flight calibration into the correction? The rest of the section does not suggest this. Or do you mean "in-flight correction" ? Please clarify. The use of the word calibration suggests calibration measurements to me, and I cannot see how that could be done in-flight. |
| SC14 | 7 | 25 | 2.3.2 | "currently" , do you expect more ghosts to become significant? |
| SC15 | 7 | 28 | 2.3.2 | Accuracy: Is this for the combined straylight or only the uniform part? I guess there must have been a discussion at some stage whether the ghost correction is useful or detrimental. You could consider adding a reference here, if there ever was any research done on this. (This is more a note of personal interest than a comment on your manuscript.) |
| SC16 | 7 | 10-22 | 2.3.2 | To make this section a bit clearer, I would change the order a bit: Lines 10-12, then lines 20 to mid 22, then lines 12 to 19, then from line 22 on. |

| # | Page | Line | Section | Comment |
|---|------|------|---------|---------|
| SC17 | 8 | 14 | 2.3.3 | Do I understand this correctly: There are two BSDF steps with a different parametrization? Why are they not combined? Is that because the BSDF depends too much on the degradation? Please clarify. |
| SC18 | 8 | 3-21 | 2.3.3 | This part might be a better read if you state what is performed for the solar, the earthshine and the moon measurements. Or is the BSDF correction also applied for earthshine and moon? Also here a figure with the processing steps and paths might help. |
| SC19 | 8 | 25-26 | 2.3.4 | I would guess that the polarization sensitivity is from on-ground calibration and that only the characterization has two main parts. This is not clear from the sentence. |
| SC20 | 8 | 31 | 2.3.4 | I assume the interpolation should be followed by a multiplication with the sensitivity? Or where does the on-ground data come back in? |
| SC21 | 9 | 9 | 2.3.4 | Why were the iterations made if they are not needed? Do you mean "for practical reasons these iterations.."? |
| SC22 | 9 | 18 | 2.4 | Can you explain which calibration parameters are included? Alternatively you could add it to the appendix. |
| SC23 | 10 | 10 | 3.0 | For the different radiometric steps a few introductory sentences at the beginning of Section 3 would be helpful. Here the approach can be outlined: that the main cause for degradation is the scan mirror, the biggest correction is done based on irradiance monitoring , then the differences between radiance and irradiance degradation are corrected in the reflectance. I found myself wondering why it was done in such a roundabout way and finding the explanation pages later. To explain the approach at the beginning of the section removes this confusion. |
| SC24 | 10 | 14 | 3.1 | I would make very clear here, that the degradation has been shown to be mainly/only the scan mirror and not the diffuser, otherwise the degradation correction doesn't seem logical. Furthermore it would be important to mention that it is a first order correction and differences between radiance and irradiance are corrected in the reflectance. |
| SC25 | 10 | 26 | 3.1 | Somewhat more explanation is needed here, why does the loss of the gyroscope functionality only affect one channel? What does this functionality do? |
| SC26 | 11 | Fig. 2 | 3.1 | From 2004 on the entire wavelength range from 450nm seems to be above 1, this is not explained in the text. Where does it come from? |
| SC27 | 11 | Tab. 1 | 3.1 | Could you also add the values for end-of-life? |
| SC28 | 11 | 4/5 | 3.1 | The same degradation is applied to both irradiance and radiance? Wouldn't that only work when all degradation occurs within the common path and none in the diffuser? Has this been verified? Then it should really be mentioned here. (OK, I now see it's mentioned later in the text. I have added SC 23 and SC24.) |
| SC29 | 12 | 13 | 3.2 | For new users of GOME data it is not clear why a platform pointing problem would only affect one channel. Please add a brief explanation. |

| # | Page | Line | Section | Comment |
|---|---|---|---|---|
| SC30 | 13 | 12/13 | 3.2 | So PMD 1 does not decay as bad as channel 2. Is it known why? Is it maybe related to the wavelength dependent difference between s-and p- reflectance of the scan mirror? |
| SC31 | 15 | 8 | 3.3 | So there is a contribution of the diffuser after all? Or is the only difference the angle on the scan mirror? |
| SC32 | 16 | 19-22 | 3.3 | Is there an explanation for the degradation getting better and worse? Have there been studies for other wavelengths too? If yes, do they show the same behaviour? |
| SC33 | 17 | Fig 6 | 3.3 | It is striking that the two wavelengths appear to behave the same until 2004 and then they start deviating, is there a reason for that? |
| SC34 | 17 | | 4.2 | This section is very well written. |
| SC35 | 19 | 2 | 4.2 | Which are the thermally sensitive optical elements? Does the degradation of thermally sensitive optical elements also cause the changes in the reflectance? |
| SC36 | 21 | Fig. 9 | 4.3.1 | The caption and plots' y-axes are not consistent with the unit and what is shown. I think you mean leakage signal or dark signal and not current. To make the plot a bit clearer, you could add the co-addition times in the plot or the caption. |
| SC37 | 21 | Fig. 9 | 4.3.1 | For channel 2 in the normal scanning mode: the spread is much larger than for the other channels and modes. Is that explained? |
| SC38 | 21 | 17 | 4.3.1 | Shouldn't it be leakage signal? |
| SC39 | 23 | 3 | 4.3.1 | Is this plot representative for other channels and modes? |
| SC40 | 25 | 4 | 4.3.2 | I assume the tape recorder failure changed the power conditioning? Or how can it have the shown effect? |
| SC41 | 26 | 5 | 4.4 | When using a monochromatic LED, the pixel response and quantum efficiency is monitored for the LED's wavelength but not necessarily for the wavelength the pixel is normally detecting. Have there ever been other measurements, for example on-ground with a white light source, to verify the results from the LEDs? |
| SC42 | 28/29 | | 5 | It's not entirely clear from the summary (or elsewhere) which insights from the long term monitoring of irradiance, degradation, spectral calibration have been included in the L1 processor. If they are, are they part of the calibration data or are corrections already included in the (ir) radiance? |
| SC43 | 29/30 | | App. A | Great idea to include the file format. |

**3. Technical corrections**

**3.1. Definitions**

Is there a reason to explicitly name the detector brand Reticon? No other brands are named as far as I could see.

Figure 11: "PDF" is not explained.

**3.2. Formatting of plots**

Figure 9: The y-axis says "DC", which normally is the dark current, but the dark signal is shown.

**3.3. Typos**

For the following words, the spelling/capitalization is not consistent throughout the article:

- Sun
- Polarization Measurement Device
- The word 'data' is used both in the singular and the plural, please pick one of the two
- Please reconsider you capitalization, either capitalize all new abbreviations or none.
  For example "Focal Plane Assembly (FPA)" but "pixel-to-pixel variations (PPG) in quantum efficiency" on page 5

| # | Page | Line | Section | Comment |
|---|---|---|---|---|
| TC1 | 1 | 2 | Abstract | Shouldn't it be "ozone *and* other trace gases"? |
| TC2 | 1 | 29 | Abstract | Shouldn't it be "polarization correction, *and* dark current correction" ? |
| TC3 | 2 | 18 | 1 | Similar changes …[], whereas *they are* |
| TC4 | 2 | 27 | 2.1 | Full stop missing. |
| TC5 | 4 | 1 | 2.1 | It's "GOME Users" not "User's" |
| TC6 | 4 | 16 | 2.1 | To clarify: "that consists *for each channel of* … " |
| TC7 | 4 | 22 | 2.1 | "… and *it had* a repeat cycle …" |
| TC8 | 4 | 22/23 | 2.1 | The sentence should also be in the past tense. |
| TC9 | 4 | 26 | 2.1 | "… additional ground stations *had* been .." |
| TC10 | 9 | 12 | 2.4 | Was thus, not thus was. |
| TC11 | 9 | 15 | 2.4 | It *has* turned out … |
| TC12 | 9 | 17 | 2.4 | Contain*s* |
| TC13 | 10 | 8 | 3.1 | Word order: the latter serve themselves… |
| TC14 | 10 | 10 | 3.1 | 3$^{rd}$ *of* July |
| TC15 | 11 | 7 | 3.1 | 3$^{rd}$ *of* July |
| TC16 | 13 | 10 | 3.1 | 3$^{rd}$ *of* July |
| TC17 | 18 | 5 | 4.2 | …lines … *have* |
| TC18 | 19 | 11 | 4.2 | The use of "however" is a bit confusing here, I would first state that they didn't find the a dependence on longitude and then "However they found the maxima… |
| TC19 | 19 | 13 | 4.2 | Temperature *rise* not raise |
| TC20 | 22 | 12 | 4.3.1 | The second "which" is not needed. |
| TC21 | 22 | 17 | 4.3.1 | The second comma is not needed. |
| TC22 | 23 | 4 | 4.3.1 | ..pixel *the* standard … |

| #    | Page | Line | Section | Comment                                                                                                                                              |
|------|------|------|---------|------------------------------------------------------------------------------------------------------------------------------------------------------|
| TC23 | 23   | 12   | 4.3.1   | Rephrase to "The most significant decrease in the number of available measurements is for the LED dark signal calibration measurements." |
| TC24 | 25   | 3    | 4.3.2   | Don't you mean the following section ?                                                                                                                |
| TC25 | 27   | 8    | 4.4     | The date format is different than before.                                                                                                            |

**3.4. References**

| Page | Line  | Section      | Comment                |
|------|-------|--------------|------------------------|
| 33   | 10/11 | Bibliography | Link seems to be faulty. |
| 33   | 12/13 | Bibliography | Link seems to be faulty. |

**3.5. Author contributions**

The authors' contributions are not listed separately, is this intentional?

---

## Referee Comment (RC2) · Anonymous Referee #2 · 18 Jun 2018

1) Scientific Significance: The manuscript provides a good description of the methods and results of a study to evaluate and improve the stability of the GOME Level 1 record.

2) Scientific Quality: The results are well-structured and well-referenced and use good statistical analysis methods. There are good references to detailed reports for interested readers.

3) Presentation Quality: The paper is well-written and the figures and tables are good in both content and structure.

Editorial Comments and Suggestions:

Make Figure 2 larger. It should at least be full page width.

In Section 2.3.2, rewrite and clarify the last line. Is this 10% the accuracy of the stray light estimates relative to the true stray light? That is, is if the stray light error is 20 units, then the correction will be between 18 and 22 units and the final result will have an error of $\pm 2$ units?

Page 19, Line 14, "raise" should be "rise".

The value of 1100 for the SNR for Channel 1 in the Table 1 seems high even for the 305 nm wavelength. What is the corresponding integration time and the size of the FOV? I believe there was a change in the Channel 1A/1B wavelength boundary during the mission. Is this before or after that change? Also, provide an SNR value for a shorter wavelength in the table, say 290 nm.

While the views of the Moon are complicated by scan mirror differences with angle and the phases of the Moon, more accurate lunar models are now available. For example, Eumetsat's GSICS Implementation of the ROLO model (GIRO) and the GSICS Lunar Observation Dataset (GLOD) introduced at

https://www.eumetsat.int/website/home/News/DAT_3460357.html?lang=EN&pState=1

could be explored to allow the lunar measurements to be used to monitor instrument changes.

Questions on Science:

Section 3.1

Page 11 line 8 et seq. While arguments can be made for estimating degradation by avoiding lines with high solar activity, this will not work well for Channel 1. See

V. Marchenko, Sergey & Deland, Matthew & Lean, Judith. (2016). Solar Spectral Irradiance Variability in Cycle 24: Observations and Models. Journal of Space Weather and Space Climate. 6. 10.1051/swsc/2016036.

for estimates of solar variability for 270 nm to 500 nm over a solar cycle. After estimating the changes in the instrument throughput, the final time-dependent solar provided in Level 1 should be constructed with realistic solar activity variations. Also, how large are the Etalon Effects in Figure 2? What errors would they be expected to produce in the radiance/irradiance ratios? Why wasn't a correction applied? It appears that the authors have access to estimates of these corrections from other analysis:

https://wdc.dlr.de/sensors/gome/degradation_files/degradation.php

And there are the earlier results in

Weber, Mark & Burrows, John & Cebula, R. (1998). GOME Solar UV/VIS Irradiance Measurements between 1995 and 1997 – First Results on Proxy Solar Activity Studies. Solar Physics. 177. 63-77. 10.1023/A:1005030909779.

Section 3.3

From Section 2.3.2, the angle for the mirror for Solar measurements is 41° and those for the Earth measurements range from 49°±15°. What are the results for Figure 6 for the ground pixels at this matching angle? If they are not equal to 1.0 what are the likely instrument changes that produce time-dependent differences in the radiance / irradiance ratios?

Is it correct that the analysis in the section is just an evaluation of errors in the Level 1 product and that no corrections based on the PICS results have been applied? If so, degradation is only shown for 325 nm and 335 nm measurements and the changes are over 20% and differ by over 5%. This does not suggest that the shorter channels are well characterized for absolute radiance / irradiance calibration. All algorithms are sensitive to the reflectance if they need parameters associated with cloud cover. What are the effects of a +10% error in the UV cloud fraction on GODFIT ozone retrievals?

Do the authors recommend that Channel 1 data in this product be used for ozone profile retrievals? What about the use of data from 300-310 for tropospheric retrievals

requiring radiance / irradiance calibration?

Section 4.2

How large are the variations in the wavelength scales along an orbit from measurement-based estimates? Do they match with the variations predicted from the effects of the measured pre-disperser prism temperature changes combined with the laboratory sensitivity characterization or are there other complicating factors? DOAS-based retrievals often generate internal estimates of the wavelength scale shifts as part of the fitting process. Have any of these been compare to this bottom-up analysis based on the prism temperatures?

---

## Author Response (AR1)

Response to Reviewer #1

We thank Reviewer #1 for her/his very detailed and helpful comments. Please find below the reviewer's comments (black) and our responses (blue) which also indicate the changes made in the manuscript.

1. General comments
The paper is well written and gives a good overview of the re-processing of GOME data together with its in-flight performance during the entire mission. The new level 1 product and new results on the instrument monitoring are definitely worth a long publication.
I find it a very good and practical idea to include a description of the new data format in the appendix, this will give the users a good starting point.
Although the overall quality of the paper content is already high, there are a number of points where the structure and phrasing can be changed to improve the overall readability.

It is not clear, which of the new in-sights and degradation corrections are included in the re-processed data set. An overview (table or graph) of the main level 1 processing steps indicating the changed steps might be an easy remedy for this. Are all the long term monitoring drift and degradation corrections included in the (ir-)radiance data now?
New users of GOME data would be helped by a few brief explanations on the mentioned GOME specifics. Especially if this new data-set is to become a reference, the paper should be as stand-alone as possible.

For the different radiometric steps a few sentences at the beginning of Section 3 would be helpful. There you could explain that the main cause for degradation is the scan mirror, the biggest correction is done based on irradiance monitoring , then the differences between radiance and irradiance degradation are corrected for in the reflectance. Explaining the approach first and then going into the details would improve the readability of this part a lot.

Section 2 of this review lists specific comments and questions about the content and understanding of the paper. These also include the issues summarized in the general comments. Once these points are clarified, I would strongly recommend the paper for publication.

Section 3 contains suggestions where to edit the text to allow for a smoother read, minor spelling and grammar errors and formatting issues.

This review is based on the version amt-2018-118.pdf retrieved on the 4th of June from https://doi.org/10.5194/amt-2018-118. The earlier version (amt-2018-118-manuscript-version1.pdf retrieved on the 1st of June) was not considered.

| # | Page | Line | Section | Comment |
|---|------|------|---------|---------|
| SC1 | 1 | 21 | 1 | You could also add S5P, S5, S4.
 Done. |
| SC2 | 3 | 12 | 2.1 | Is the integration time for the forward scan also 1.5 s? This is not clear from the text.
 Yes. We made this more clear in the text. |
| SC3 | 4 | 12 | 2.1 | What kind of channel separator? In line 14 a di-chroic is mentioned, is a di-chroic used here too?
 It is a channel separator prism. We added an explanation. |
| SC4 | 5 | 5 | 2.2 | Please add which parameters are calibrated in-flight and which are based on on-ground data only.
 Done. In the list we indicate the in-flight calibration parameters with an asterisk. |
| SC5 | 5 | 7-17 | 2.2 | It would be very useful to have a flowchart showing the actual order of |

| | | | | the steps, also indicating which steps have been changed/improved with the new version. Maybe even show which step is based on in-flight data. The word "steps" in line 7 suggests that the list shows the order of the steps. Consider writing "the basic calibration algorithms are:" instead. We added two more figures (Figs. 2 and 3) indicating the processing flow for calculating the in-flight calibration parameters and the science data, respectively. The description of the algorithms using the on-ground calibration has been moved into a separate section (new Sec. 3). |
|---|---|---|---|---|
| SC6 | 6 | 8/9/10 | 2.2 | Are these calibration constants then fed back immediately into the L1 processor? Yes, they are fed back immediately. We added this information to the text. |
| SC7 | 6 | 9 | 2.2 | The use of the word " comprise" is confusing here. It suggests that the list is complete, that only data from dark measurements, the PtNeCr lamp and the LED is used to directly derive calibration constants during L1 processing. Is this indeed correct? I would also expect that for example transient filtering is performed for all data during L1 processing. Yes, this list is complete. Transient filtering is not performed. No changes were made in the text. |
| SC8 | 6 | 30 | 2.3.1 | What are the other sources? What is "slow" noise? "Slow" with respect to what? "Additional" is not needed here. In this case "slow" means "the noise varies slowly with time and not from readout to readout". We reformulated the sentence. |
| SC9 | 6 | 32 | 2.3.1 | "from one typical orbit": Isn't this section about on-ground calibration data? Does this scaling factor ever change? This scaling factor has been obtained from one orbit during the commissioning phase. It has not changed since then. |
| SC10 | 6 | 33 | 2.3.1 | So only band 1a earthshine needs this correction as all other bands and modes have a shorter integration time? Yes. |
| SC11 | 7 | 2 | 2.3.1 | Are all Peltier signals involved for all detectors? Or only the Peltier belonging to band 1 a? Please clarify. Only the Peltier output belonging to channel 1 is used. We added the information in the text. |
| SC12 | 7 | 6/7 | 2.3.1 | So the additional residual correction is not part of the L1 processor discussed here? Was it not feasible to include this? The correction is implemented in the L1 processor. We clarified this in the text. |
| SC13 | 7 | 12 | 2.3.2 | "In-flight calibration exercise" : do you actually include in-flight calibration into the correction? The rest of the section does not suggest this. Or do you mean "in-flight correction" ? Please clarify. The use of the word calibration suggests calibration measurements to me, and I cannot see how that could be done in-flight. We replaced "in-flight calibration exercise" with "L1 processing". |
| SC14 | 7 | 25 | 2.3.2 | "currently" , do you expect more ghosts to become significant? We deleted "currently". |
| SC15 | 7 | 28 | 2.3.2 | Accuracy: Is this for the combined straylight or only the uniform part? I |

| | | | | guess there must have been a discussion at some stage whether the ghost correction is useful or detrimental. You could consider adding a reference here, if there ever was any research done on this. (This is more a note of personal interest than a comment on your manuscript.)
It is for the uniform part. Unfortunately, there is no reference available. It has been discussed pre-flight (1994), but we do not have more detailed information. |
|---|---|---|---|---|
| SC16 | 7 | 10-22 | 2.3.2 | To make this section a bit clearer, I would change the order a bit: Lines 10-12, then lines 20 to mid 22, then lines 12 to 19, then from line 22 on.
Done as suggested. |
| SC17 | 8 | 14 | 2.3.3 | Do I understand this correctly: There are two BSDF steps with a different parametrization? Why are they not combined? Is that because the BSDF depends too much on the degradation? Please clarify.
Yes, there are two BSDF steps. This has historical reasons. In the previous GDP version the application of a separate extraction software was required in which the second step was performed. The second BSDF step contains an improved azimuth dependence. We made this more clear in the text. |
| SC18 | 8 | 3-21 | 2.3.3 | This part might be a better read if you state what is performed for the solar, the earthshine and the moon measurements. Or is the BSDF correction also applied for earthshine and moon? Also here a figure with the processing steps and paths might help.
This section (now Sec. 3.3) has been rewritten to make this more clear. |
| SC19 | 8 | 25-26 | 2.3.4 | I would guess that the polarization sensitivity is from on-ground calibration and that only the characterization has two main parts. This is not clear from the sentence.
We split the sentence in order to make this more clear. |
| SC20 | 8 | 31 | 2.3.4 | I assume the interpolation should be followed by a multiplication with the sensitivity? Or where does the on-ground data come back in?
We added the information in line 26. |
| SC21 | 9 | 9 | 2.3.4 | Why were the iterations made if they are not needed? Do you mean "for practical reasons these iterations.."?
The iterations were made in the course of several (unplanned) reprocessings. We added this in the text. |
| SC22 | 9 | 18 | 2.4 | Can you explain which calibration parameters are included? Alternatively you could add it to the appendix.
In the appendix we added the reference to the Product User Manual (Aberle, 2018) that contains the complete list of all parameters. |
| SC23 | 10 | 10 | 3.0 | For the different radiometric steps a few introductory sentences at the beginning of Section 3 would be helpful. Here the approach can be outlined: that the main cause for degradation is the scan mirror, the biggest correction is done based on irradiance monitoring , then the differences between radiance and irradiance degradation are corrected in the reflectance. I found myself wondering why it was done in such a roundabout way and finding the explanation pages later. To explain the approach at the beginning of the section removes this confusion.
We added a brief introduction to this section (now Sec. 4). |

| SC24 | 10 | 14 | 3.1 | 1) I would make very clear here, that the degradation has been shown to be mainly/only the scan mirror and not the diffuser, otherwise the degradation correction doesn't seem logical.
2) Furthermore it would be important to mention that it is a first order correction and differences between radiance and irradiance are corrected in the reflectance.
1) We added a sentence and a reference here to make this more clear.
2) We mention this in Section 'Reflectance Degradation' (former Sec. 3.3) |
|------|----|----|-----|----------------------------------------------------------------------|
| SC25 | 10 | 26 | 3.1 | Somewhat more explanation is needed here, why does the loss of the gyroscope functionality only affect one channel? What does this functionality do?
Our text was a bit misleading in this case. The loss of the gyroscope functionality did affect all channels. We reordered the sentences to make this more clear. |
| SC26 | 11 | Fig. 2 | 3.1 | From 2004 on the entire wavelength range from 450nm seems to be above 1, this is not explained in the text. Where does it come from?
Values above 1 might be due deposits on the coatings which can lead to changes in interference patterns and to an increase in intensity (Snel, 2001). We added the explanation in the text. |
| SC27 | 11 | Tab. 1 | 3.1 | Could you also add the values for end-of-life?
End-of-life values have been added (and also values for 290nm, see comments Reviewer #2). |
| SC28 | 11 | 4/5 | 3.1 | The same degradation is applied to both irradiance and radiance? Wouldn't that only work when all degradation occurs within the common path and none in the diffuser? Has this been verified? Then it should really be mentioned here. (OK, I now see it's mentioned later in the text. I have added SC 23 and SC24.)
Please see responses to SC23 and SC24. |
| SC29 | 12 | 13 | 3.2 | For new users of GOME data it is not clear why a platform pointing problem would only affect one channel. Please add a brief explanation.
See response to SC25. We reordered the sentences here, too. |
| SC30 | 13 | 12/13 | 3.2 | So PMD 1 does not decay as bad as channel 2. Is it known why? Is it maybe related to the wavelength dependent difference between s-and p- reflectance of the scan mirror?
We assume that this might be related to long-term changes in the mean wavelength of the PMD which is in the order of ~10nm for PMD1. |
| SC31 | 15 | 8 | 3.3 | So there is a contribution of the diffuser after all? Or is the only difference the angle on the scan mirror?
The only difference is the angle on the scan mirror. |
| SC32 | 16 | 19-22 | 3.3 | Is there an explanation for the degradation getting better and worse? Have there been studies for other wavelengths too? If yes, do they show the same behaviour?
We did not analyze other wavelengths. Degradation getting better and worse might be due to changes in interference patterns (Snel, 2001). We added a sentence and the reference. |
| SC33 | 17 | Fig 6 | 3.3 | It is striking that the two wavelengths appear to behave the same until 2004 and then they start deviating, is there a reason for that? |

| | | | | We assume that this is related to the scan angle dependence that increased over time (see Snel, 2001). |
|---|---|---|---|---|
| SC34 | 17 | | 4.2 | This section is very well written. |
| SC35 | 19 | 2 | 4.2 | Which are the thermally sensitive optical elements? Does the degradation of thermally sensitive optical elements also cause the changes in the reflectance?
We removed this sentence. |
| SC36 | 21 | Fig. 9 | 4.3.1 | The caption and plots' y-axes are not consistent with the unit and what is shown. I think you mean leakage signal or dark signal and not current. To make the plot a bit clearer, you could add the co-addition times in the plot or the caption.
Corrected. |
| SC37 | 21 | Fig. 9 | 4.3.1 | For channel 2 in the normal scanning mode: the spread is much larger than for the other channels and modes. Is that explained?
Unfortunately, we do not have an explanation for this behavior. |
| SC38 | 21 | 17 | 4.3.1 | Shouldn't it be leakage signal?
Yes. Corrected. |
| SC39 | 23 | 3 | 4.3.1 | Is this plot representative for other channels and modes?
Yes, a three-fold broadening of the distribution was also found for other bands and modes. |
| SC40 | 25 | 4 | 4.3.2 | I assume the tape recorder failure changed the power conditioning? Or how can it have the shown effect?
The noise level changed because of the significantly reduced number of measurements and in particular because measurements from the South Atlantic Anomaly region are missing since the tape recorder failure. |
| SC41 | 26 | 5 | 4.4 | When using a monochromatic LED, the pixel response and quantum efficiency is monitored for the LED's wavelength but not necessarily for the wavelength the pixel is normally detecting. Have there ever been other measurements, for example on-ground with a white light source, to verify the results from the LEDs?
Unfortunately, this has not been verified. |
| SC42 | 28/29 | | 5 | It's not entirely clear from the summary (or elsewhere) which insights from the long term monitoring of irradiance, degradation, spectral calibration have been included in the L1 processor. If they are, are they part of the calibration data or are corrections already included in the (ir) radiance?
More results of the study were included in the summary in order to make this more clear. |
| SC43 | 29/30 | | App. A | Great idea to include the file format.
Thanks. |

**3. Technical corrections**

**3.1. Definitions**

Is there a reason to explicitly name the detector brand Reticon? No other brands are named as far as I could see.

No, there is no explicit reason, it is rather a 'leftover'. We replaced it with "array detector".

Figure 11: "PDF" is not explained.
We added the explanation.

**3.2. Formatting of plots**

Figure 9: The y-axis says "DC", which normally is the dark current, but the dark signal is shown.
Corrected.

**3.3. Typos**

For the following words, the spelling/capitalization is not consistent throughout the article:
- Sun
- Polarization Measurement Device
- The word 'data' is used both in the singular and the plural, please pick one of the two
- Please reconsider you capitalization, either capitalize all new abbreviations or none. For example "Focal Plane Assembly (FPA)" but "pixel-to-pixel variations (PPG) in quantum efficiency" on page 5

Spelling/capitalization should be consistent now.

| # | Page | Line | Section | Comment |
|---|---|---|---|---|
| TC1 | 1 | 2 | Abstract | Shouldn't it be "ozone and other trace gases"?
 Changed. |
| TC2 | 1 | 29 | Abstract | Shouldn't it be "polarization correction, and dark current correction" ?
 Changed. |
| TC3 | 2 | 18 | 1 | Similar changes ...[], whereas they are
 Corrected. |
| TC4 | 2 3 | 27 | 2.1 | Full stop missing.
 Inserted. |
| TC5 | 4 | 1 | 2.1 | It's "GOME Users" not "User's"
 Corrected. |
| TC6 | 4 | 16 | 2.1 | To clarify: "that consists for each channel of ... "
 Added. |
| TC7 | 4 | 22 | 2.1 | "... and it had a repeat cycle ..."
 Added. |
| TC8 | 4 | 22/23 | 2.1 | The sentence should also be in the past tense.
 Changed. |
| TC9 | 4 | 26 | 2.1 | "... additional ground stations had been .."
 Changed. |
| TC10 | 9 | 12 | 2.4 | Was thus, not thus was.
 Corrected. |
| TC11 | 9 | 15 | 2.4 | It has turned out ...
 Corrected. |
| TC12 | 9 | 17 | 2.4 | Contains
 Corrected. |
| TC13 | 10 | 8 | 3.1 | Word order: the latter serve themselves...
 Changed. |
| TC14 | 10 | 10 | 3.1 | 3rd of July |

| | | | | Corrected. |
|---|---|---|---|---|
| TC15 | 11 | 7 | 3.1 | 3rd of July
Corrected. |
| TC16 | 13 | 10 | 3.1 | 3rd of July
Corrected. |
| TC17 | 18 | 5 | 4.2 | ...lines ... have
Corrected. |
| TC18 | 19 | 11 | 4.2 | The use of "however" is a bit confusing here, I would first state that they didn't find a dependence on longitude and then "However they found the maxima...
Reformulated. |
| TC19 | 19 | 13 | 4.2 | Temperature rise not raise
Corrected. |
| TC20 | 22 | 12 | 4.3.1 | The second "which" is not needed.
Changed. |
| TC21 | 22 | 17 | 4.3.1 | The second comma is not needed.
Changed. |
| TC22 | 23 | 4 | 4.3.1 | ..pixel the standard ...
Changed. |
| TC23 | 23 | 12 | 4.3.1 | Rephrase to "The most significant decrease in the number of available measurements is for the LED dark signal calibration measurements."
Changed. |
| TC24 | 25 | 3 | 4.3.2 | Don't you mean the following section ?
Corrected. |
| TC25 | 27 | 8 | 4.4 | The date format is different than before.
Changed. |

**3.4. References**

| Page | Line | Section | Comment |
|---|---|---|---|
| 33 | 10/11 | Bibliography | Link seems to be faulty.
Corrected. |
| 33 | 12/13 | Bibliography | Link seems to be faulty.
We double-checked the link and it should be valid. |

**3.5. Author contributions**

The authors' contributions are not listed separately, is this intentional?
This section is optional and we would like to leave it out.

Response to Reviewer #2

We thank Reviewer #2 for her/his very detailed and helpful comments. Please find below the reviewer's comments (black) and our responses (blue) which also indicate the changes made in the manuscript.

1) Scientific Significance: The manuscript provides a good description of the methods and results of a study to evaluate and improve the stability of the GOME Level 1 record.

2) Scientific Quality: The results are well-structured and well-referenced and use good statistical analysis methods. There are good references to detailed reports for interested readers.

3) Presentation Quality: The paper is well-written and the figures and tables are good in both content and structure.

Editorial Comments and Suggestions:

Make Figure 2 larger. It should at least be full page width.
Done as suggested.

In Section 2.3.2, rewrite and clarify the last line. Is this 10% the accuracy of the stray light estimates relative to the true stray light? That is, is if the stray light error is 20 units, then the correction will be between 18 and 22 units and the final result will have an error of ±2 units?
We added: "… not more accurate than ~10%, i.e. processing errors of 10% of true straylight."

Page 19, Line 14, "raise" should be "rise".
Corrected.

The value of 1100 for the SNR for Channel 1 in the Table 1 seems high even for the 305 nm wavelength. What is the corresponding integration time and the size of the FOV? I believe there was a change in the Channel 1A/1B wavelength boundary during the mission. Is this before or after that change? Also, provide an SNR value for a shorter wavelength in the table, say 290 nm.
We added to Table 1:
  •   the information on the integration time for channel 1 (6s);
  •   values for 290nm;
  •   values for middle (2001) and end (2010) of the mission.
The change in the Channel 1A/1B wavelength boundary was implemented in June 1998.

While the views of the Moon are complicated by scan mirror differences with angle and the phases of the Moon, more accurate lunar models are now available. For example, Eumetsat's GSICS Implementation of the ROLO model (GIRO) and the GSICS Lunar Observation Dataset (GLOD) introduced at https://www.eumetsat.int/website/home/News/DAT_3460357.html?lang=EN&pState=1 could be explored to allow the lunar measurements to be used to monitor instrument changes.
Thank you very much for the reference to these data sets! For the GOME instrument one problem is that the Moon measurements do not fill the entire slit so that the calibration key data cannot be applied just like that. Moreover, the Moon is always observed at a scan angle in the western Limb, whereas scan angle dependency observations corresponding to East or Nadir pixels would be needed to significantly improve on the degradation correction using the Sun. And since the Moon merely reflects Sun light, it cannot be used as calibration source independent of solar activity.

Questions on Science:

Section 3.1
Page 11 line 8 et seq. While arguments can be made for estimating degradation by avoiding lines with high solar activity, this will not work well for Channel 1. See
V. Marchenko, Sergey & Deland, Matthew & Lean, Judith. (2016). Solar Spectral
Irradiance Variability in Cycle 24: Observations and Models. Journal of Space Weather
and Space Climate. 6. 10.1051/swsc/2016036.
for estimates of solar variability for 270 nm to 500 nm over a solar cycle. After estimating the changes in the instrument throughput, the final time-dependent solar provided in Level 1 should be constructed with realistic solar activity variations.

We agree with the reviewer that in principle realistic solar variations should be taken into account. Thank you for pointing us to this reference. Nonetheless, providing an optimum solar irradiance product is not the main focus of this study.

Also, how large are the Etalon Effects in Figure 2? What errors would they be expected to produce in the radiance/irradiance ratios? Why wasn't a correction applied? It appears that the authors have access to estimates of these corrections from other analysis:
https://wdc.dlr.de/sensors/gome/degradation_files/degradation.php

The etalon amplitudes may be estimated from Figure 4 (former Fig. 2) as the amplitude of the semi-regular wiggles (~10 wiggles in channel 1 to ~5 in channel 4). Since GOME does not have a flat field mode (e.g. using a white light source) etalon cannot be directly derived. Correction using the Sun would be possible but only relative to a certain reference date, not in an absolute sense. The main focus of the Level 1 product has been to function as input for Level 2 retrievals. For Level 2, etalon is irrelevant as long as the structure is identical for solar and for earth-shine measurements. There are in fact some indications that this may not completely be the case, depending on the solar azimuth, but attempts to characterize solar-azimuth dependent etalon-like structures of the diffuser BSDF have not been deemed reliable enough to be applied in the GOME calibration. The errors are shown in Fig.8 of reference (Slijkhuis 2004) on
https://wdc.dlr.de/sensors/gome/degradation_files/degradation.php

And there are the earlier results in
Weber, Mark & Burrows, John & Cebula, R. (1998). GOME Solar UV/VIS Irradiance Measurements between 1995 and 1997 – First Results on Proxy Solar Activity Studies. Solar Physics. 177. 63-77. 10.1023/A:1005030909779.

Section 3.3

From Section 2.3.2, the angle for the mirror for Solar measurements is 41◦ and those for the Earth measurements range from 49◦ ±15◦. What are the results for Figure 6 for the ground pixels at this matching angle? If they are not equal to 1.0 what are the likely instrument changes that produce time-dependent differences in the radiance/irradiance ratios?

In Fig. 8 (former Fig. 6) the results for the west pixels (incidence angle 44°-34°), cyan curves, match the angle for the solar measurements. In general, west pixels show the minimal degradation in reflectance compared to the other ground pixels types. Explicit characterization and indication of the instrument changes that produce the reflectance degradation is difficult. As mentioned on page 10: "The main degradation as a consequence of extensive exposure to the space environment can be attributed to deposits on the scan mirror (which is coated with a MgF 2 layer) thereby changing its reflective properties". This change in mirror coating also changes the scan-angle dependent polarization properties of the instrument (Snel, 2001) .

Is it correct that the analysis in the section is just an evaluation of errors in the Level 1 product and that no

corrections based on the PICS results have been applied?
Yes, this is correct. We added this to the summary.

If so, degradation is only shown for 325 nm and 335 nm measurements and the changes are over 20% and differ by over 5%. This does not suggest that the shorter channel are well characterized for absolute radiance / irradiance calibration. All algorithms are sensitive to the reflectance if they need parameters associated with cloud cover. What are the effects of a +10% error in the UV cloud fraction on GODFIT ozone retrievals?
A 10% error in cloud fraction is expected to have an impact of <1% on total column ozone retrieval (Christophe Lerot, personal communication, July 2018).

Do the authors recommend that Channel 1 data in this product be used for ozone profile retrievals? What about the use of data from 300-310 for tropospheric retrievals requiring radiance / irradiance calibration?
We agree with the reviewer that the retrieval of ozone profiles and tropospheric columns from GOME requires a very careful handling of the measured spectra and additional corrections to account for degradation. However, several studies successfully demonstrated the feasibility (Liu et al., 2005, Cai et al., 2012, Miles et al., 2015). Moreover, Keppens et al. (2018) have shown that decadal drift values for GOME level-2 ozone profiles are overall insignificant.

Section 4.2

How large are the variations in the wavelength scales along an orbit from measurement-based estimates? Do they match with the variations predicted from the effects of the measured pre-disperser prism temperature changes combined with the laboratory sensitivity characterization or are there other complicating factors?
On average 9 different wavelength scales are used along one orbit. The variation in the wavelengths depends on the spectral region. In general, the variation is <0.002 nm, except for the beginning of channel 3 and the end of channel 4, where the variation is 0.004-0.005 nm along one orbit. This analysis is based on ~2000 randomly selected orbits.
The use of spectral calibration as function of pre-disperser temperature was a recommendation based on on-ground measurements of instrument performance. However, we cannot retrieve the original data.

DOAS- based retrievals often generate internal estimates of the wavelength scale shifts as part of the fitting process. Have any of these been compare to this bottom-up analysis based on the prism temperatures?
No, unfortunately, these comparisons have not been performed.

[revised manuscript text omitted]

The paper is organized as follows: In Section 2 we provide an overview of the GOME instrument design  and brief descriptions of the level-0-to-1  processing chain and the new level 1 product. Section  3 contains 
[revised manuscript text omitted]
 (in-flight calibration parameters) are derived during the level 0-to-1 processing. They are fed back immediately to the processor. This comprises the dark signal measurements on the night side of each orbit, the internal LED measurements, and at regular intervals wavelength calibration using the spectral lamp measurements. Figure 2 depicts the processing flow for calculating the respective in-flight calibration parameters including the solar reference measurements. The calibration parameters as well as the Sun Mean Reference (SMR) spectrum are stored in the calibration data base. Monitoring these calibration parameters provides an excellent insight into the long-term stability of the instrument. A detailed description of the corresponding algorithms and the results of the long-term analysis is presented in Sect. 5.

Figure 3 is a flowchart indicating the order of the steps for processing and calculating the level 1 science data after the calculation of the calibration data. The individual algorithms are applied to the pre-processed solar data, moon and earthshine measurements. 'Normalize' means the normalization of the signal to 1 second exposure time. Detailed descriptions of the individual algorithms are presented in Sec. 3 for the on-ground calibration, and in Sec. 5 for the in-flight calibration. Another step in the entire calibration procedure is the correction of degradation (see Sect. 4.1). Due to degradation in optical components the calibration  parameters for radiance and irradiance change in time. However, this degradation cannot be derived from on-board calibration sources and the correction has to be obtained offline and externally from the data processor. For GOME this has been done by scientific analysis of the solar observations.

[Figure]

**Figure 2.** Processing flow for calculating the in-flight calibration parameters from the dark signal measurements (DARK), the internal LED measurements (LED), the spectral lamp measurements (LAMP), and the solar measurements (SUN). The calibration parameters as well as the Sun Mean Reference (SMR) spectrum are stored in the calibration data base.

[Figure]

**Figure 3.** Processing flow indicating the order of steps for calculating the calibrated level 1 science data, i.e. irradiance (left column), moon radiance (middle column), and earthshine radiance (right columns). See text for detailed explanations.

[revised manuscript text omitted]
 is probably not more accurate than ∼10%, i.e. processing errors of 10% of true stray light

**3.2.1 Radiometric calibration**

**3.3 Radiometric calibration**

The objective of the radiometric calibration is to transform the 16-bit Binary Units (BU) of the detector pixel readouts into calibrated radiances (photons s$^{-1}$ cm$^{-2}$ nm$^{-1}$ sr$^{-1}$) or, for the Sun, into calibrated irradiance (photons s$^{-1}$ cm$^{-2}$ nm$^{-1}$).

In GDP-L1 the radiometric calibration is divided into several steps (see also Fig. 3).

The radiance response function, which depends on wavelength, scan angle, and temperature, is applied to the solar, moon, and earthshine measurements. It is a compound function in which the scan angle dependent part and the temperature dependent part are given per channel, for 9 scan angles and for 5 temperatures, respectively. These key data are then interpolated to the actual values of the respective measurement. Then, solar and earthshine spectra are corrected for instrument degradation (see Sect. 4.1).

The BSDF correction is applied to the solar measurements and comprises two parts. The basic BSDF from the on-ground calibration depends on wavelength, azimuth angle, and the elevation of the sunlight on the diffuser. It is expressed as parametrization using polynomials. The second step uses an improved azimuth dependence of the diffuser BSDF (Slijkhuis et al., 2006). The azimuth dependence is fitted using a third-order polynomial in wavelength for all channels. The polynomial coefficients are stored in a look-up-table for a number of azimuth angles which are then linearly interpolated to the actual angle.

The earthshine radiance is additionally corrected for the so-called 'radiance jump' effect that is caused by the serial readout of the detector, i.e. the last pixel of the array is read out 93.75 ms later than the first pixel. In case of

inhomogeneous ground scenes this effect may be visible as a jump in radiance between two neighboring detectors. The last pixels of one detector record the same wavelengths as the first pixels of the next channel, but at an integration time shifted by 93.75 ms. A linear correction in wavelength is applied which re-normalizes all intensities to the same integration time thereby using information from the PMDs (which are read out every 93.75 ms synchronized with the first detector pixel). Although the correction adjusts the continuum level, it cannot account for any difference in spectral features that may arise from viewing a slightly different ground pixel. For earthshine measurements the intensity calibration also includes the application of a polarization correction (see Sect. 3.4).

**3.3.1 Polarization correction**

**3.4 Polarization correction**

GOME is a polarization sensitive instrument. The radiance response function described in Sect.  3.3 calibrates the instrument assuming unpolarized light. Therefore a correction factor must be applied which describes the ratio

$$c_{pol} = \frac{\text{throughput\_for\_actual\_input\_polarization}}{\text{throughput\_for\_unpolarized\_light}} .$$

[revised manuscript text omitted]

small. Values above 1 might be due deposits on the coatings which can lead to changes in interference patterns and an increase in intensity (Snel, 2001) . Since mid-2001 
[revised manuscript text omitted]

The polarization correction algorithm was improved in the new GDP-L1 5.1. Instead of climatological values the ozone columns derived from the GOME measurements themselves are used for the parametrization of the Generalized Distribution Function. By means of the daily solar irradiance measurements the degradation was monitored and corrected. Degradation can be explained in terms of deposits on the GOME scan mirror. Below 300 nm intensity decreased by 80-95% which implies a significant deterioration of the signal-to-noise ratio and which may have a severe impact on the challenging retrieval of atmospheric parameters such as ozone profiles. The decrease in channel 2 is 40-80%. In channel 3 the decrease (10-40%) started in 2001, whereas throughput changes in channel4 are relatively small. Since 2001 the measurements were additionally affected by an ERS-2 pointing problem.  A correction algorithm has been developed and further improved which relies on the intensity measured in the early part of the mission and which comprises a wavelength- and a time-dependent part. In GDP-L1 this correction is routinely applied to irradiance and radiance measurements. The degradation in reflectance, i.e. the differential degradation between solar irradiance and earth radiance measurements has been monitored for two wavelengths, 325 and 335 nm (lower and upper limit of the ozone fitting window), using cloud-free pixels over the Saharan desert. Changes are in the order of -10 to 30% and depend on wavelength and the viewing angle. Since changes in reflectance may result from both changes in instrument performance or changes in atmospheric conditions, no routine corrections are applied in GDP-L1.

For the spectral calibration special attention was paid to the identification of lamp lines that remain stable (with respect to the statistical moments) over the whole mission. This has resulted in an updated spectral line list that improved the temporal stability of the wavelength assignment. For the leakage current an increase of 4 BU/s per decade  and a widening of the distribution were found. Typically, in GDP-L1 dark signal measurements from the same  or a very close-by orbit are applied so that these changes do not have a negative impact on the measurement quality. The existing dark signal correction has been further improved by differentiating between measurements from outside and inside the SAA. Thereby, the enhanced background signal and noise level, which are typical for measurements from inside the SAA, are better accounted for. The output of the LEDs that are used to monitor the pixel-to-pixel sensitivity decreased to about 60% of the early-mission values. For channel 2 a significant broadening of the PPG distribution was observed.

*Data availability.* The new GOME L1 products can be accessed via https://earth.esa.int/web/guest/news/-/asset_publisher/G2mU/content/new-ers-2-gome-level-1-v5-1-dataset-available-online.

**Appendix A: NetCDF structure of new level 1 files**

The GOME level 1 product filename is constructed as follows:

      <MMM>_<CCCC>_<TTTTTTTTTT>_.nc,

where <MMM> is the mission ID, <CCCC> is the file class, and <TTTTTTTTTT> (= <FFFF><DDDDDD>) is a mission specific file type. <FFFF> is the file category and <DDDDDD> is a product semantic descriptor.  consists of start time, end time, orbit number, packet version, processor version, and processing time. The packet version is a version number which is specific for the combination of processor version, input data (for example calibration data) version and configuration version. For GOME Level 1 products, the mission ID is ER2 for ERS-2. The file class can be TEST  or RPRO

for test data or reprocessing . The file type field contains an instrument identifier (GOM) as file category and the processing level (L1B_ or L2__ or L0__) as semantic descriptor. The packet version is 2 and the processor version is currently 5.1. We encode these versions into "02SUBSCRIPTNB051000" as the versions part of the "instance ID". The file extension is ".nc" that is typically used for netCDF files. All time strings in the filename and product are formatted in ISO 6801 format. Following this scheme, the result would be for example: ER2_TEST_GOM_L1B____20010811T032404_20010811T050712_32981_02_051000_20150311T151024.nc . Product size may vary between 60 and 75MB. Products which are measured after the  ERS-2 tape recorder problem in June 2003 are typically smaller because they don't  comprise measurements for the entire orbit.

The different dimensions in the GOME level 1 netCDF file are time (=1), which corresponds to  one time per orbit, scanline (≈500), which corresponds to one complete scan comprising three forward and one backward scan, ground_pixel (3 or 1), which corresponds to the number of across-track scans, detector channel (=4), which corresponds to the number of detectors, band (=6), which corresponds to the number of spectral windows, and spectral_channel, which corresponds to the total number of detector pixels.

Figure A1 provides an overview of the netCDF structure of the level 1 file. In addition to metadata and instrument related parameters, calibration data and irradiance measurements are available. The radiance measurements themselves are organized in groups for different modes : nadir, static_view, narrow_swath, north- and southpolar_view, sun, or moon. Forward and backward scans are separated in different groups. Inside these groups there are subgroups for bands and PMDs. A band is a part of a channel which can have its own integration time and co-adding factor. Integration times may change during one orbit. All subgroups contain several variables and attributes. For a detailed description and the complete list of all variables we refer to Aberle (2018).

*Competing interests.* The authors declare that they have no conflicts of interest

*Acknowledgements.* This work was performed in the framework of the ESA GOME-Evolution project. Particular thanks to Wolfgang Lengert, ERS-2 Mission Manager, for making this work possible. Thanks to Kai-Uwe Eichmann (IUP-B), Andreas Richter (IUP-B), Mark Weber (IUP-B), Steffen Beirle (MPIC), and Gabriele Brizzi (SERCO) for helpful feedback on the prototype version of the new GOME L1 product.

[Figure]

**Figure A1.** NetCDF structure of new GOME level 1 file version 5.1. Measurements are organized in groups for different modes and bands (see text for more explanations).